# Macroscopic weavable fibers of carbon nanotubes with giant thermoelectric power factor

Natsumi Komatsu [1,2], Yota Ichinose[3], Oliver S. Dewey[2,4], Lauren W. Taylor[2,4], Mitchell A. Trafford[2,4], Yohei Yomogida[3], Geoff Wehmeyer[2,5], Matteo Pasquali [2,4,6,7], Kazuhiro Yanagi [3] & Junichiro Kono [1,2,7,8✉]

Low-dimensional materials have recently attracted much interest as thermoelectric materials because of their charge carrier confinement leading to thermoelectric performance enhancement. Carbon nanotubes are promising candidates because of their one-dimensionality in addition to their unique advantages such as flexibility and light weight. However, preserving the large power factor of individual carbon nanotubes in macroscopic assemblies has been challenging, primarily due to poor sample morphology and a lack of proper Fermi energy tuning. Here, we report an ultrahigh value of power factor ($14 \pm 5 \, \text{mW m}^{-1} \text{K}^{-2}$) for macroscopic weavable fibers of aligned carbon nanotubes with ultrahigh electrical and thermal conductivity. The observed giant power factor originates from the ultrahigh electrical conductivity achieved through excellent sample morphology, combined with an enhanced Seebeck coefficient through Fermi energy tuning. We fabricate a textile thermoelectric generator based on these carbon nanotube fibers, which demonstrates high thermoelectric performance, weavability, and scalability. The giant power factor we observe make these fibers strong candidates for the emerging field of thermoelectric active cooling, which requires a large thermoelectric power factor and a large thermal conductivity at the same time.

[1] Department of Electrical and Computer Engineering, Rice University, Houston, TX, USA. [2] Carbon Hub, Rice University, Houston, TX, USA. [3] Department of Physics, Tokyo Metropolitan University, Tokyo, Japan. [4] Department of Chemical and Biomolecular Engineering, Rice University, Houston, TX, USA. [5] Department of Mechanical Engineering, Rice University, Houston, TX, USA. [6] Department of Chemistry, Rice University, Houston, TX, USA. [7] Department of Materials Science and NanoEngineering, Rice University, Houston, TX, USA. [8] Department of Physics and Astronomy, Rice University, Houston, TX, USA. ✉email: kono@rice.edu

Thermoelectric (TE) materials convert heat into electricity and vice versa, offering great potential for waste heat recovery and solid-state cooling[1]. TE materials are usually evaluated by the $ZT$ factor, defined as $ZT = S^2\sigma T\kappa^{-1}$, where $S$ is the Seebeck coefficient, $\sigma$ is the electrical conductivity, $\kappa$ is the thermal conductivity, and $T$ is the temperature. While previous studies on thermoelectric materials have primarily focused on reducing $\kappa$ to improve $ZT$, enhancing the power factor ($PF$), defined as $PF = S^2\sigma$, is more important for certain applications. For example, for the energy harvesting application, large $PF$ is crucial for maximizing the output power density when the heat source is unlimited (such as solar heat and industrial waste heat)[2–4]. Furthermore, $PF$ must be large for so-called active cooling[5,6], in which the Peltier effect is leveraged to enhance the heat flow rates from the hot side to the ambient temperature. This active cooling mode is promising for electronics thermal management applications, and is distinct from the more traditional refrigeration operational mode in which heat is pumped from the cold side to the hot side via the Peltier effect[5,6]. The maximum hot-side heat flow rate in active cooling is proportional to the effective thermal conductivity $\kappa_{\text{eff}}$, defined as[5] $\kappa_{\text{eff}} = \kappa + \frac{PF \cdot T_H^2}{2\Delta T}$, where $T_H$ is the hot-side temperature and $\Delta T$ is the temperature difference between the two sides, suggesting that the active cooling requires large $\kappa$ together with large $PF$, instead of high $ZT$.

In addition to the basic TE properties, practical applications require other considerations, such as toxicity, flexibility, and scalability[7,8]. Conventional inorganic TE materials such as $Bi_2Te_3$ and their alloys have shown high performance, e.g., $ZT \sim 1.2$ and $PF \sim 4.5$ mW m$^{-1}$ K$^{-2}$ at room temperature[9]. However, their toxicity, scarcity, and rigidity prevent their wide use. On the other hand, organic materials are safe, flexible, and inexpensive, but they have exhibited small $PF$ values[7]. These issues have resulted in a search for organic-like materials with inorganic-like TE performance.

Low-dimensional materials are believed to hold the key to achieving this goal. Recent studies have reported record-high $PF$ values at room temperature for two-dimensional (2D) materials: monolayer graphene [36.6 mW m$^{-1}$ K$^{-2}$ at 290 K[10]] and ultra-thin FeSe [26 mW m$^{-1}$ K$^{-2}$ at 280 K[11]]. However, such demonstrations have been limited to small flakes, and whether they are scalable for practical applications is questionable. One-dimensional (1D) materials such as carbon nanotubes (CNTs) are expected to possess even better TE properties[12]. 1D quantum confinement of charge carriers leads to enhanced TE performance through a narrow carrier distribution achieved when the Fermi energy, $E_F$, is near a 1D van Hove singularity (VHS) in the density of states (DOS)[13]. Furthermore, in recent years, significant improvements have been made in fabricating macroscopically ordered CNT assemblies[14] with superb thermal[15,16] and mechanical properties[17], suggesting that CNTs can be used for creating an ideal TE material with high TE performance, flexibility, and scalability simultaneously.

Single-wall CNTs (SWCNTs) can be either semiconducting or metallic, depending on their chirality, $(n, m)$[18]. For TE device applications, mainly semiconducting SWCNTs have been studied because of their larger $S$ compared to metallic SWCNTs[19,20]. As in any semiconductor, however, the maximum $S$ is achieved when $E_F$ is near the charge neutrality point (CNP) in the middle of the bandgap, where $\sigma$ (and thus $PF$) is negligibly small because the DOS is zero. An attempt to increase $\sigma$ by moving $E_F$ toward a band-edge decreases $S$; this trade-off between $S$ and $\sigma$ is a well-known dilemma for TE material development[21]. Recently, Ichinose et al. have experimentally demonstrated that metallic SWCNTs can show higher $PF$ than semiconducting SWCNTs when $E_F$ is in the vicinity of a VHS through simultaneous

enhancement of $\sigma$ and $S$[22]. The same scenario holds for a mixture of semiconducting and metallic SWCNTs, and a theoretical study[23] predicts a $PF$ higher than 100 mW m$^{-1}$ K$^{-2}$. However, experimentally measured $PF$ for CNT assemblies has remained small[8], presumably due to low $\sigma$ originating from poor sample morphology.

Here, we studied the TE properties of macroscopic weavable CNT fibers. These neat CNT fibers simultaneously possess a high degree of CNT alignment, a high density, a high CNT aspect ratio (length per diameter), and a low level of impurities[16], leading to ultrahigh electrical conductivity[15], $\sigma > 10$ MS m$^{-1}$. We tuned $E_F$ to the vicinity of a 1D VHS through a chemical treatment to maximize $S$, obtaining $PF$ as high as $14 \pm 5$ mW m$^{-1}$ K$^{-2}$. This is the highest $PF$ value achieved for any CNT system and approaching the highest values reported for 2D materials[10,11]. We developed a theoretical model to explain the $E_F$ dependence of $PF$ and validated it with finer $E_F$ tuning using electrolyte gating. Finally, we demonstrated weavability and scalability by fabricating a textile TE generator based on these CNT fibers, which produced enough power to turn on a light-emitting diode (LED).

## Results and discussion

**Giant power factor in ultrahigh-conductivity CNT fibers.** CNTs used to fabricate fibers mainly contained double-wall CNTs (DWCNTs) with an average outer (inner) wall diameter of $1.8 \pm 0.2$ nm ($0.9 \pm 0.1$ nm), measured by high-resolution transmission electron microscopy (HR-TEM) (Supplementary Fig. 1). The viscosity-averaged aspect ratio was $6.7(\pm 0.1) \times 10^3$, measured using a capillary thinning extensional rheometer[24]. A solution spinning method[15,16] was used to spin CNTs into a continuous fiber. CNTs were first dissolved in chlorosulfonic acid (CSA) to create a spin dope[25]. The dope was then filtered and extruded into a coagulant. Finally, the coagulated fiber was collected onto a rotating drum. This method produced meters (>100 m) of fiber with densely packed and highly aligned CNTs, as shown in Fig. 1a. The average diameter of the fiber was determined to be $8.9 \pm 0.9$ μm by scanning electron microscopy (SEM). Dimensions of CNTs and the produced CNT fiber are summarized in Supplementary Table 1. Raman spectroscopy was performed on the produced CNT fibers (Supplementary Fig. 2b, c). The average G-to-D ratio was 50 (with 532 nm excitation), demonstrating a low density of defects (Supplementary Fig. 2b). The radial breathing mode (RBM) region of the Raman spectra indicates the presence of CNTs with diameters ranging from 0.8 to 2.0 nm (Supplementary Fig. 2c), consistent with results from HR-TEM measurements. The CNTs inside the solution spun fiber have a high aspect ratio, as well as a low impurity density, and are highly crystalline, leading to exceptional mechanical (a tensile strength of $4.2 \pm 0.2$ GPa) and electrical ($\sigma > 10$ MS m$^{-1}$) properties while retaining flexibility[15,17].

We chemically treated as-produced CNT fibers to tune $E_F$ (conditions summarized in Supplementary Table 2). The as-produced CNT fibers were heavily p-doped with CSA during the solution spinning process, and exhibited $\sigma$ of $11 \pm 2$ MS m$^{-1}$ (measured independently in three laboratories). Doping an as-produced CNT fiber with iodine monochloride (ICl) increased the value of $\sigma$ to $16 \pm 3$ MS m$^{-1}$ through further p-type doping, while annealing them at a temperature of 350 °C (500 °C) decreased $\sigma$ to $5.6 \pm 1.1$ MS m$^{-1}$ ($2.7 \pm 0.5$ MS m$^{-1}$) through dedoping. Further characterization of the annealed fibers is described in Supplementary Fig. 3.

We measured the $\sigma$ and $S$ of these CNT fibers at room temperature under vacuum using an experimental setup schematically shown in Fig. 1b; see "Methods" section for more details about the measurements. The measured $S$ values were all positive,

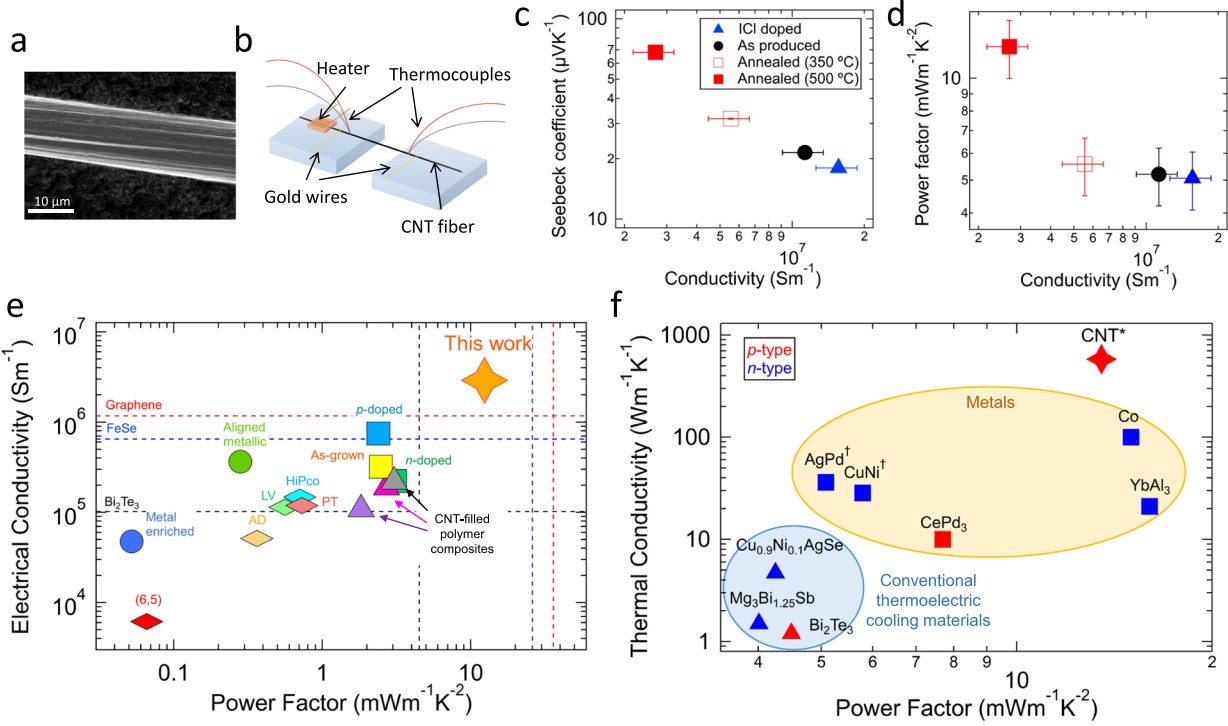

**Fig. 1 Thermoelectric properties of densely packed and highly aligned carbon nanotube (CNT) fibers. a** SEM image of a CNT fiber. **b** Schematic of the experimental setup used for measuring the electrical conductivity ($\sigma$) and Seebeck coefficient ($S$) of CNT fibers. **c** Measured $S$ and **d** corresponding power factor ($PF = S^2\sigma$) as a function of $\sigma$ for four CNT fibers that underwent different chemical treatments: Iodine monochloride (ICl) doped (solid blue triangles), as-produced (solid black circles), annealed at 350 °C (open red squares), and annealed at 500 °C (solid red squares). The values are summarized in Supplementary Table 2. Error bars indicate standard deviation (SD). **e** Comparison of reported $PF$ values for various CNT samples with $\sigma$. The plot includes unsorted CNTs (as-grown (yellow square)[44], p-doped (blue square)[45] and n-doped (green square)[26]), semiconductor-enriched SWNCTs (originally produced by arc-discharge (AD) (orange diamond)[46], by laser vaporization (LV) (green diamond)[46], by HiPco (blue diamond)[46], by plasma torch (PT) (pink diamond)[46], and (6,5) SWCNTs (red diamond)[22]), metal-enriched SWCNTs (un-aligned films (blue circle)[22] and aligned films (green circle)[22]), and CNT-filled polymer nanocomposites (PANI/graphene/PANI/DWCNT (purple triangle)[47], PANI/graphene-PEDOT:PSS/PANI/DWCNT-PEDOT:PSS (pink triangle)[48], and PANI/graphene/PANI/DWCNT (gray triangle)[27]). Values are summarized in Supplementary Table 4. $PF$ values of Bi$_2$Te$_3$ alloys[9], graphene[10], and FeSe[11] serve as references. **f** Comparison of reported $PF$ values for representative materials with the thermal conductivity ($\kappa$) at 300 K with a temperature difference ($\Delta T$) of 1 K. The plot includes metals (squares) (Co[49]), (YbAl$_3$[50], CePd$_3$[51], CuNi[52], and AgPd[52]) and conventional thermoelectric cooling materials (triangles) (Bi$_2$Te$_3$[9], Cu$_{0.9}$Ni$_{0.1}$AgSe[53], and Mg$_3$Bi$_{1.25}$Sb[54]). p-type (n-type) materials are highlighted in red (blue). †The temperature was at 400 K because values at 300 K were not available. *Thermal conductivity value was taken from ref.[16]. Values are summarized in Supplementary Table 5.

indicating that the carrier type was p-type in these samples. Figure 1c shows a monotonic decrease of $S$ with increasing $\sigma$, resulting in a decrease of $PF$ with $\sigma$ (Fig. 1d). The highest $S$ was obtained for the CNT fiber annealed at 500 °C, and the measurements were repeated for three samples to ensure reproducibility. The average $S$ of the three samples (all annealed at 500 °C) was $68.0 \pm 0.3$ µV K$^{-1}$, corresponding to an average (maximum) $PF$ of $12 \pm 2$ mW m$^{-1}$ K$^{-2}$ ($14 \pm 5$ mW m$^{-1}$ K$^{-2}$). Assuming the thermal conductivity for similar annealed CNT fibers[16], 580 W m$^{-1}$ K$^{-1}$, the average (maximum) $ZT$ is estimated to be $6 \times 10^{-3}$ ($7 \times 10^{-3}$) at 300 K.

This maximum $PF$ value, $14 \pm 5$ mW m$^{-1}$ K$^{-2}$, is the highest value ever achieved for any CNT sample. Figure 1e summarizes the room-temperature $PF$ values reported for different CNT systems with $\sigma$. The highest $PF$ among CNT systems has been ~3 mW m$^{-1}$ K$^{-2}$, achieved for an unsorted benzyl viologen doped CNT web[26] and CNT-filled polymer nanocomposites[27]. Furthermore, our value is over three times larger than that of Bi$_2$Te$_3$, the commercially used inorganic p-type TE material (~4.5 mW m$^{-1}$ K$^{-2}$)[9], and is approaching the highest $PF$ achieved at room temperature by 2D materials: monolayer graphene (36.6 mW m$^{-1}$ K$^{-2}$)[10]. Note that

the giant $PF$ in this work was observed in macroscopic samples (~1 cm length), whereas the highest $PF$ reported for 2D materials has been measured in microscopic samples (typically in µm-scale). CNT fibers were produced in continuous runs (over 100 m in total length, as discussed above) and cut into centimeter-scale pieces to conduct measurements. We further demonstrate this strength, i.e., scalability, in a later section by using CNT threads that were produced by plying multiple fibers via a continuous plying machine.

Furthermore, the high $PF$ observed is promising for use in active cooling, leveraging the high thermal conductivity (580 W m$^{-1}$ K$^{-1}$) of the CNT fibers[16]. TE active cooling requires a material with large $\kappa$ and large $PF$, simultaneously, to maximize $\kappa_{\text{eff}}$. However, no existing TE materials satisfy this requirement, as shown in Fig. 1f. Conventional TE cooling materials have relatively large $PF$, but small $\kappa$. Adams et al. used a magnon-drag metal (Co) and a Kondo-effect metal (CePd$_3$), which showed $\kappa_{\text{eff}}$ of 780 W m$^{-1}$ K$^{-1}$ and 360 W m$^{-1}$ K$^{-1}$ at 300 K with a $\Delta T$ of 1 K, respectively[5]. Given the same temperature difference ($\Delta T = 1$ K), the $\kappa_{\text{eff}}$ of our CNT fibers is expected to be 1190 W m$^{-1}$ K$^{-1}$, exceeding those of Co and CePd$_3$ as well as conventional TE materials[28] (Supplementary Table 5).

**Theoretical model to explain $E_F$ dependence of TE properties.**
We first estimated the $E_F$ of the four chemically treated fibers discussed in Fig. 1 by conducting systematic optical spectroscopy measurements. Thin films of aligned CNTs produced by a facile blade coating technique[29] were used, because their size and absorption were more suited for optical measurements. This method also starts from dissolving CNTs in CSA as in the solution spinning method, and then produces films instead of fibers. We used the same raw CNT materials and ensured that the chemical treatments (e.g., CSA concentration, doping, and annealing conditions) were identical to the treatments to the fibers discussed above. Supplementary Fig. 7a compares expected absorbance peak positions based on the diameter distribution[30,31] determined by the TEM analysis with absorbance spectra for the four films, annealed at 500 °C, annealed at 350 °C, as-produced, and ICl doped, respectively. We estimated $E_F$ through optical absorption spectral analysis. For example, the peak at ~0.57 eV is due to the $E_{11}$ transition of the outer semiconducting CNTs; it is visible in the annealed samples but is suppressed in the as-produced and ICl doped samples. The suppression of the peak suggests that $E_F$ resides inside the valence band, causing Pauli blocking. By analyzing other peaks in the same manner, the $E_F$ of the annealed (as-produced and doped) samples was estimated to be in the vicinity of the first VHS of the outer (inner) semiconducting tubes (Supplementary Fig. 7b). More details are discussed in Supplementary Note 1. $\sigma$, $S$, and $PF$ measured in Fig. 1 are plotted as a function of the estimated $E_F$ in Fig. 2a–c, respectively.

To understand the $E_F$ dependence of $S$ and $PF$ (Fig. 1c, d, respectively), we developed a theoretical model and performed simulations. We chose four representative SWCNTs with appropriate diameters to describe the DWCNT fibers: an inner-wall semiconducting SWCNT (SC1), an inner-wall metallic SWCNT (M1), an outer-wall semiconducting SWCNT (SC2), and an outer-wall metallic SWCNT (M2); see Supplementary Table 3. We first calculated the DOS (Fig. 2d), $S_{ind}$, electrical conductance $G_{ind}$, and $PF'_{ind} \equiv G_{ind}S_{ind}^2$ for each individual SWCNT. Next, we modeled a DWCNT as consisting of two individual SWCNTs corresponding to the inner and outer nanotubes, while adopting circuit models[23,32,33] to approximate our DWCNT fiber. We obtained the combined conductance $G_p$ ($G_s$) and Seebeck coefficient $S_p$ ($S_s$) for the parallel (series) case. Then we further combined $S_p$ and $S_s$ through $S_{tot} = (1 - \beta)S_s + \beta S_p$[19], where $\beta$ is the fraction of the parallel component. We set $\beta$ to be 0.9 and assumed $G_{tot} = G_p$ to best fit the experimental data. More details about the calculation methods are given in Supplementary Note 2.

Calculated $G_{tot}$, $S_{tot}$, and $PF'_{tot} \equiv G_{tot}S_{tot}^2$ using the combined model as a function of $E_F$ are shown in Fig. 2a–c, respectively. Their $E_F$ dependence based on the combined model qualitatively differs from those based on individual semiconducting SWCNTs such as SC1 and SC2 (Supplementary Fig. 8). As shown in Fig. 2a, $G_{tot}$ is finite even when the $E_F$ is inside the bandgap. Moreover, maximum $|S_{tot}|$ appears when $E_F$ is in the vicinity of the first VHS of SC2, not when $E_F$ is near the CNP, i.e., $E_F = 0$ eV (Fig. 2b). This is because the peak of $|S_{ind}|$ near the CNP, expected for SC1 or SC2 alone, is suppressed in our combined system due to their nearly zero conductance inside the bandgap (Supplementary Eq. 9). Figure 2b, c demonstrates that $S_{tot}$ and $PF'_{tot}$ show a maximum value when the $E_F$ is around the first VHS of SC2, and the next peak appears when the $E_F$ is around the second VHS of SC2 (overlapping with the first VHS of SC1), consistent with previous studies[22,23]. Figure 2a–c also compares calculation results (left axis) with the experimental data (right axis) for $\sigma$, $S$, and $PF$, respectively. Experimental data and

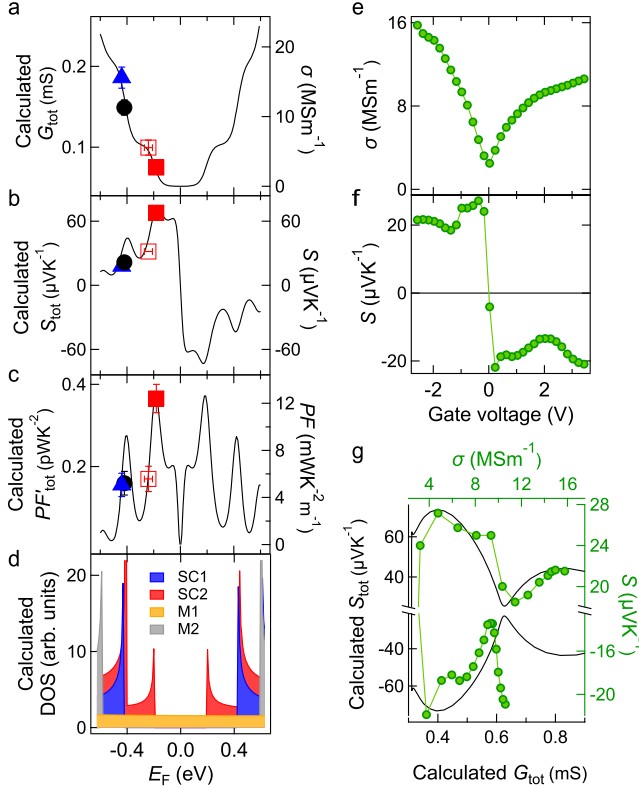

**Fig. 2 Explaining the $E_F$-dependent thermoelectric (TE) properties of carbon nanotube (CNT) fibers through modeling. a** Calculated electrical conductance $G_{tot}$, **b** Seebeck coefficient $S_{tot}$, and **c** power factor $PF'_{tot} \equiv G_{tot}S_{tot}^2$ (left) versus Fermi energy $E_F$. Experimental data from four chemically treated samples are shown by different symbols on the right axis as a function of the estimated Fermi energy: iodine monochloride (ICl) doped (solid blue triangle), as-produced (solid black circle), annealed at 350 °C (open red square), and annealed at 500 °C (solid red square). Error bars indicate SD. **d** The density of states (DOS) of four representative SWCNTs contained in our CNT fibers, an inner-wall semiconducting SWCNT (SC1, blue), an inner-wall metallic SWCNT (M1, orange), an outer-wall semiconducting SWCNT (SC2, red), and an outer-wall metallic SWCNT (M2, gray). **e, f** TE properties tuned by the electrolyte gating technique. **e,** Measured electrical conductivity $\sigma$ and **f** $S$ as a function of gate voltage. **g** Comparison of calculated $S_{tot}$ (left) as a function of $G_{tot}$ (bottom) with experimental $S$ (right) as a function of $\sigma$ (top) with an offset in the x axis such that the examined range by the experiments agrees. The calculated $S_{tot}$ is plotted with a black line, and the experimental data for gated samples are shown by a green line with solid circles.

calculated values demonstrated excellent qualitative agreement, proving that the highest value of $PF$ observed for the CNT fibers is a result of $E_F$ being tuned to the vicinity of the VHS.

To validate these calculations, we further measured the $\sigma$, $S$, and $PF$ of a CNT fiber while varying $E_F$ using the electrolyte gating technique[22,34]. Specifically, we injected electrons or holes into the CNT fiber to shift the $E_F$ by changing the gate voltage $V_G$. The CNT fiber was annealed at 500 °C before the measurements to remove residual dopants. Figure 2e, f shows the measured $\sigma$ and $S$ as a function of $V_G$, respectively. In both plots, the CNP corresponds to $V_G = 0$; i.e., the experimental $V_G$ values were shifted such that $\sigma$ becomes minimum at $V_G = 0$. The observation of both positive and negative $S$ and a transistor-like behavior of $\sigma$ indicate that $E_F$ was tuned from the p-type regime to the n-type regime. The $S$ values presented in Fig. 2f are lower than those in Fig. 1c because the sample was immersed in the ionic liquid.

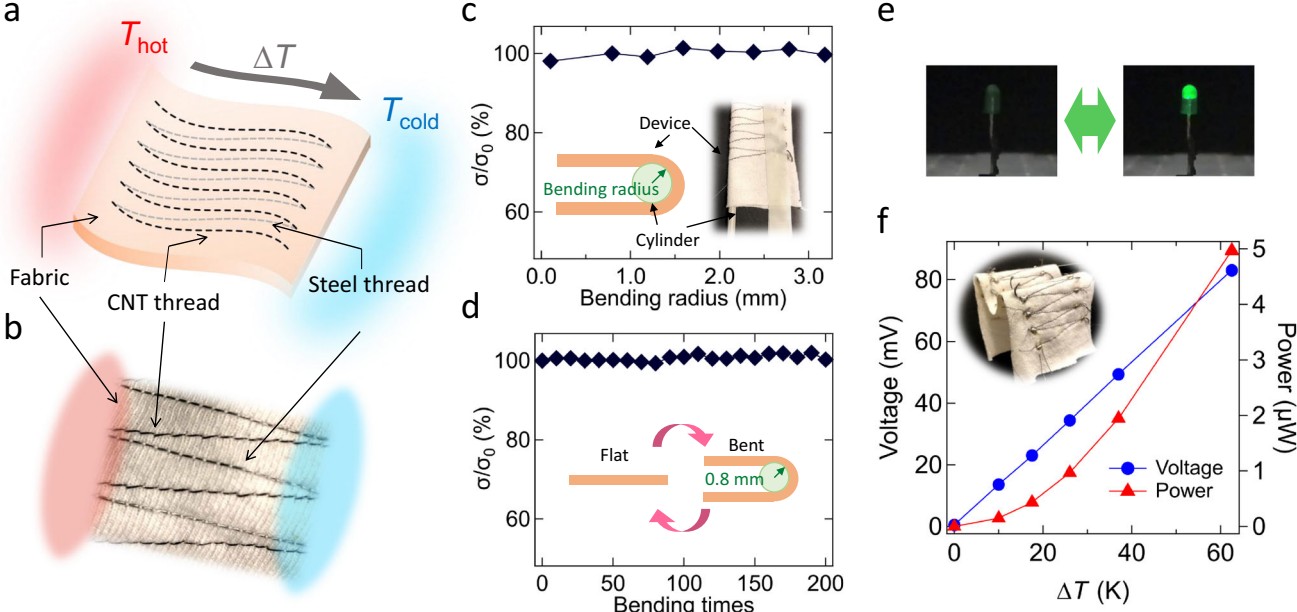

**Fig. 3 Textile thermoelectric (TE) generator based on carbon nanotube (CNT) threads sewn into fabric. a** Schematic and **b** photograph of the device. CNT threads (p-type thermoelectric generator) and steel threads (for electrical connection) were sewn into a fabric using a sewing machine. The CNT threads were connected electrically in series and thermally in parallel. A temperature difference $\Delta T$ was applied by heating one side ($T_{hot}$) while keeping the other side ($T_{cold}$) at room temperature. **c** Electrical conductivity $\sigma$ of one TEG unit as a function of bending radius, normalized by the conductivity without bending ($\sigma_0$). The inset shows a schematic and a picture of the device during the measurement. **d** $\sigma$ as a function of bending number, normalized by that of the original state ($\sigma_0$). The bending radius was 0.8 mm. The inset shows a schematic of the measurement procedure. **e** The entire device was connected to an LED through an amplification circuit and a capacitor. The LED turned on with a $\Delta T$ of ~50 K. **f** Output voltage and power as a function of applied $\Delta T$. The inset shows a folded device, demonstrating its flexibility.

Figure 2g plots the calculated $S_{tot}$ (experimental $S$) as a function of calculated $G_{tot}$ (experimental $\sigma$) for direct comparison of our calculations with the gate-dependent data. The overall behavior ($S$ increases, decreases, and then increases again with $\sigma$) is well reproduced, validating our model.

**Origin of the giant power factor**. The observed giant $PF$ can be explained as a combined effect of excellent sample morphology, which led to the ultrahigh $\sigma$, and the ability to tune $E_F$ to the vicinity of a VHS. To compare the observed TE properties of our CNT fibers with those of previous CNT samples, Supplementary Fig. 17a, b shows the maximum values of $S$ and $PF$, respectively, in each sample as a function of $\sigma$. The maximum value of $S$ in this study is comparable to $S$ values reported in previous studies (Supplementary Fig. 17a), while the maximum $PF$ value in this study is distinctly higher than other $PF$ values (Supplementary Fig. 17b), suggesting that the ultrahigh conductivity was key to the achieved giant $PF$. Although $\sigma$ for individual metallic SWCNTs has been reported[35,36] to be ~100 MS m$^{-1}$, $\sigma$ for CNT assemblies used in TE studies has been lower than 1 MS m$^{-1}$. In this study, the excellent morphology of the aligned CNT fibers led to the well-preserved $\sigma$ of >10 MS m$^{-1}$ even in a macroscopic assembly, resulting in the ultrahigh value of $PF$. $PF$ values for CNT assemblies can be further enhanced in the future by improving the sample morphology, better preserving the $\sigma$ of individual CNTs.

In addition to improving $\sigma$, tuning $E_F$ is also crucial for maximizing $PF$. Supplementary Fig. 17c, d shows not only the maximum values but also other experimental values from $E_F$ tuning measurements for this study and for samples from ref. [22]. As shown in Supplementary Fig. 17d, within the same sample, increasing $\sigma$ does not always result in improving $PF$; in this study, increasing $\sigma$ decreased both $S$ and $PF$. Furthermore, the decrease

or increase of $S$ and $PF$ with respect to $\sigma$ is not monotonic; $S$, as well as $PF$, show a peak when $E_F$ is near a VHS in the DOS, as demonstrated in Fig. 2. Therefore, it is important to understand the $E_F$ dependence of TE properties, and tune $E_F$ accordingly to maximize $PF$.

These results highlight the promising properties of CNTs as a TE material. Conventionally, materials with high $\sigma$ were considered undesirable for TE device applications because of the well-known trade-off between $S$ and $\sigma$. However, as shown in this study, CNTs can provide relatively large $S$ if $E_F$ is properly tuned, in spite of having an ultrahigh $\sigma$.

**Textile TE generator based on macroscopic weavable CNT threads**. The high $PF$ value that we found, in addition to their weavability and scalability, make CNT fibers promising building blocks for the emerging technology of fiber and textile electronics[37–40]. CNT fibers are not only weavable by using a commercial sewing machine but also washable[41]. These unique properties are difficult to achieve using other functional fibers. Here, we fabricated a high-performance textile TE generator based on CNT threads sewn into fabric. The CNT thread was produced by plying 21 CNT filaments together. The average diameter of the CNT thread was 190 μm. Dimensions for CNTs and CNT threads are summarized in Supplementary Table 1. We sewed the CNT threads and stainless steel threads into a fabric (100% cotton) using a sewing machine such that the CNT threads were connected electrically in series and thermally in parallel, as shown in Fig. 3a, b. We generated a temperature difference, $\Delta T$, across the device by heating one side ($T_{hot}$) with hotplates while keeping the other side ($T_{cold}$) at room temperature. The CNT threads generated power through the Seebeck effect, while the steel threads provided electrical connections.

We investigated the flexibility of this device by performing bending tests. The device was wound around cylinders with specific diameters, as shown in the inset of Fig. 3c, and the electrical conductivity ($\sigma$) at each bending radius was compared to that of the original state without bending ($\sigma_0$). Figure 3c shows that no significant change (<2%) occurred up to a bending radius of 0.1 mm. We repeated the bending for 200 times with a fixed bending radius of 0.8 mm. As shown in Fig. 3d, the conductivity did not change more than 1.9%. Furthermore, we applied a temperature difference to the device while it was bent. Supplementary Fig. 18e shows a generated voltage as a function of the temperature difference during bending, verifying that there was essentially no degradation in device performance due to bending.

Using this CNT-based textile TE generator, we demonstrated powering of a LED. The entire device consisted of sixty CNT threads connected, and we utilized RC circuits and a DC/DC converter to drive the LED, as shown in Supplementary Fig. 20. We first charged the capacitors with the textile TE generator, and then discharged them to turn on the LED (see also Supplementary Movie 1). With a $\Delta T$ of >50 K, the textile TE generator provided a high enough voltage to light up the LED after the conversion, as shown in Fig. 3e. Figure 3f demonstrates the output voltage and power of the device with 60 CNT threads. The device generated a voltage of 83 mV and a power of 5.0 μW when a $\Delta T$ of 60 K was applied. Further improvements on the device architecture can be made to achieve out-of-plane energy harvesting and cooling, which are promising for wearable devices[42,43].

In summary, we studied the thermoelectric properties of aligned CNT fibers with an ultrahigh electrical conductivity ( > $10^7$ S m$^{-1}$) by varying the Fermi energy and obtained a maximum power factor of $14 \pm 5$ mW m$^{-1}$ K$^{-2}$. This ultrahigh value of power factor was achieved by a combined effect of the ultrahigh electrical conductivity and the ability to tune Fermi energy to the vicinity of a van Hove singularity. Our theoretical study, combined with gating experiments, validated the Fermi energy dependence of the observed power factor, and suggested that it can be further improved by electrical conductivity enhancement. We demonstrated the weavability and scalability of the CNT fibers by fabricating a textile thermoelectric generator sewn into fabric. The coexistence of giant thermoelectric power factor, flexibility, and scalability in the CNT fibers promise a diverse range of TE applications, including the thermoelectric active cooling application, where both a large power factor and a large thermal conductivity are required.

## Methods

**Solution spinning method**. CNTs were produced by Meijo Nano Carbon Co. and high-resolution transmission electron microscopy (HR-TEM, JEOL 2100F operating at 200 kV) was performed on the raw CNT material (Supplementary Fig. 1). As-received carbon nanotubes were dispersed in acetone using light tip sonication for two minutes and dropped onto a Lacey carbon grid and allowed to dry in air. The average outer-wall diameter was $1.8 \pm 0.2$ nm, the average inner-wall diameter was $0.9 \pm 0.1$ nm, and the average number of walls was 1.9. The viscosity-averaged aspect ratio of the CNTs was $6700 \pm 100$, as measured with a Trimaster capillary thinning extensional rheometer[24].

A solution spinning method[15,16] was used to spin CNTs into a continuous fiber. CNT fibers were spun from chlorosulfonic acid (CSA) (Sigma-Aldrich, 99%), coagulated into acetone, and collected onto a rotating drum. Unlike other dispersion techniques such as ultrasonication and chemical functionalization, this method for dispersion does not induce defects in the $sp^2$ bonding of the CNTs and does not shorten the CNTs. Supplementary Fig. 2a shows a representative scanning electron microscopy (SEM) (FEI Helios NanoLab 660 DualBeam) image of the surface of the produced CNT fiber, used to measure the average diameter. The average diameter was $8.9 \pm 0.9$ μm, and the total length was 60 m, which can be further extended by a longer spinning time.

Raman spectroscopy (Renishaw InVia Confocal Raman microscope) was used to characterize produced CNT fibers. The tensile strength was measured with an

ARES G2 rheometer (TA Instruments). Fibers had a gauge length of 20 mm and were broken at a speed of 0.1 mm s$^{-1}$.

**Chemical treatment to solution spun fibers**. Doping: Iodine monochloride (ICl) was introduced to the fiber structure via vapor-phase doping (evaporate dopant in presence of CNT fiber). The fiber was gently wound around a glass vial and secured at each end with Kapton tape to prevent tangling. The dopant was handled in a dry-air glove box to minimize interactions with moisture in the air. The dopant and fibers were combined at room temperature in a round-bottom flask (RBF) purged with nitrogen gas. The RBF was sealed and placed under vacuum until the dopant began to evaporate and was placed in a box oven at the doping temperature. The box oven was masked with aluminum foil to minimize light exposure. After doping, the RBF was removed from the oven and chilled in a water bath followed by an ice water bath to encourage dopant condensation and crystallization at the inner wall of the RBF instead of on the CNT fiber. The fiber was immediately removed from the glass vial and subsequently taped in a folder for storage.

Annealing: The CNT fibers were annealed in the furnace (ARF-30K, ASH (Asahi Rikagaku)) under vacuum (~mTorr) with a temperature controller (AGC-S, ASH (Asahi Rikagaku)) at the target temperature (350 or 500 °C) for 4 h.

**Textile thermoelectric generator (TEG) fabrication**. The raw CNT material used for the device was produced by Meijo Nano Carbon Co., and the average diameter was determined to be 1.8 nm with an average of 1.5 walls. The aspect ratio was 4100. CNT fibers were produced using the same method but using a 2 wt% solution of CNTs in CSA. The fibers were washed in water, dried in oven overnight to stabilize electrical properties for textile applications. Twenty-one CNT filaments were plied together to create a sewing thread, and the average diameter of the thread was 190 μm.

The CNT thread and stainless steel thread (NGW-1pc Conductive Stainless Steel Sewing Thread), were sewn onto the fabric in 4 cm lengths using a commercial sewing machine (Singer 2277 Tradition). Silver paste (PELCO® Conductive Silver Paint, Ted Pella, Inc.) was applied at junctions for better electrical connections. Fifteen CNT threads were connected thermally in parallel and electrically in series to create one TEG unit, and the four units were connected in series as shown in Supplementary Figs. 19 and 20. The resistance of the entire TEG was ~300 Ω. The cold side was at room temperature, and the hot side was placed on the hotplates.

**Thermoelectric measurement (without electrolyte gating)**. The CNT fiber was suspended between two glass slides. The channel length was 0.6–1.2 cm. A heater (KFR-02N-120-C1-11N10C2, Kyowa Dengyo Co.) was attached on one side of the fiber, and thermocouples (KFT(TW)-50-100-050, ANBE SMT Co.) were fixed on the fiber at the edge of glass slides by silver paste (D-500 DOTITE, Fujikura Kasei Co.). Gold wires were attached to the fiber, next to the thermocouples, by silver paste. A device picture is shown in Supplementary Fig. 4a. Electrical conductivity and Seebeck coefficient measurements were conducted in a vacuum (~$10^{-3}$ Pa) using a vacuum and low-temperature probe station (Grail 10, Nagase Techno Co.). The conductivity was calculated from the measured resistance of the fiber. The Seebeck coefficient was measured by applying a temperature difference by the heater and measuring the generated voltage, in the same manner as that described in ref. [34]. We used three different contact materials, silver (D-500 DOTITE, Fujikura Kasei Co.), carbon (DOTITE XC-12, Nisshin EM Co.), and gold (No. 8560, Tokuriki Honten Co., Ltd.), to ensure the reproducibility of Seebeck coefficient.

**Thermoelectric measurement (with electrolyte gating)**. The CNT fiber was transferred onto a glass slide with pre-deposited gold electrodes (thickness ~ 100 nm). A heater, thermocouples and gold wires were fixed on the fiber as described above. To ensure that no chemical reaction occurs between silver and ionic liquid, insulating sealant (TSE397-C, Momentive Performance Materials Japan LLC.) covered the silver paste. The ionic liquid (TMPA-TFSI, Kanto Chemical Co.) was dropped to cover the CNT fiber and gate electrodes to create an electrolyte gating system. A device picture is shown in Supplementary Fig. 4b. Electrical conductivity and Seebeck coefficient measurements were conducted in the same system as above. By changing the gate voltage from +3.6 V to −3.2 V, we injected electrons or holes into the CNT fiber, shifting the $E_F$. For the transport measurements, the source-drain voltage was kept as small as possible (3 mV), and then the transport properties were evaluated in the linear response region. The Seebeck coefficients were always measured in the same gate-shift direction, such as from positive to negative, in order to eliminate the influence of hysteresis during the measurements. An offset applied to the experimental $V_G$ values such that $\sigma$ becomes minimum at $V_G = 0$ was 0.5 V. Note that the offset value depends on the initial doping level of CNT samples.

**Powering a light-emitting diode (LED) demonstration**. The entire circuit to power an LED is shown in Supplementary Figs. 19 and 20. It consists of four TEG units connected electrically in series, five capacitors connected in parallel (total capacitance of 16.5 mF), an LED (TEG-DMO, Custom Thermoelectric, LLC), and a single pole double throw switch. The LED is attached to a uni-polar boost converter circuit (VB0410-1, TXL Group, Inc.) to convert low voltage to higher values.

The converter operates when the input voltage is 40 mV or higher, and the output voltage is 1–10 V depending on the input voltage and load. When the switch is connected to A, the capacitor is charged by the TEGs through $Q(t) = CV_S[1 - \exp(-t/R_SC)]$, where $Q$ is the charge, $C$ is the capacitance, $V_S$ is the voltage generated by the TEGs, $t$ is time, and $R_S$ is the resistance of the TEGs. When it switches to B, the capacitor discharges through $Q(t) = CV_C \exp(-t/R_LC)$, where $V_C$ is the voltage drop across the capacitor at $t = 0$ (thus $V_C = V_S$) and $R_L$ is the resistance of the converter, to drive the LED (Supplementary Movie 1). Note that the converter was solely operated by the TEGs and no additional power was required.

**Textile TEG output voltage and power measurement.** The cold side of the flexible TEG (four TEG units) was at room temperature, and the hot side was placed on hotplates, which were monitored by the temperature controller (Model 340 Temperature Controller, Lake Shore Cryotronics, Inc.). The target temperature was set on the hotplate, and we waited until the temperature stabilized. We first directly connected the TEG to the voltmeter (Series 2400 Source Measure Unit, Keithley) to measure the open-circuit voltage (Fig. 3f). We then connected the TEG to a load resistor, and measured the voltage drop across the load resistor as well as the current through the resistor. Supplementary Fig. 21 shows the output voltage and power for multiple temperature differences. Figure 3f plots the maximum power, which was obtained when the load resistor value was the same as the TEG resistance (~300 Ω).

## Data availability

All other data that support the plots within this paper and other findings of this study are available from the corresponding authors upon reasonable request. Source data are provided with this paper.

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

## Acknowledgements

N.K. and J.K. acknowledge support by the Basic Energy Science (BES) program of the U. S. Department of Energy through Grant No. DE-FG02-06ER46308 (for preparation of aligned carbon nanotube films), the U.S. National Science Foundation through Grant No. ECCS-1708315 (for optical measurements), and the Robert A. Welch Foundation through Grant No. C-1509 (for structural characterization measurements). K.Y. acknowledges support by JSPS KAKENHI through Grant Numbers JP17H06124, JP17H01069, JP18H01816, and JP20H02573 and the JST CREST program through Grant Number JPMJCR17I5, Japan. Y.I. acknowledges support by JSPS KAKENHI through Grant Number JP19J21142, Japan. O.S.D., L.W.T., M.T., and M.P. acknowledge support by the U.S. Air Force research through Grant No. FA9550-15-1-0370, the Robert A. Welch Foundation through Grant C-1668, and the Department of Energy awards DE-EE0007865 (Office of Energy Efficiency and Renewable Energy - Advanced Manufacturing Office) and DE-AR0001015 (Advanced Research Projects Agency - Energy). L. W.T. was supported by the Department of Defense through a National Defense Science and Engineering Graduate (NDSEG) Fellowship, 32 CFR 168a. O.S.D. was partially supported by a Riki Kobayashi Fellowship from the Rice Chemical & Biomolecular Engineering Department. N.K., O.S.D., L.W.T., M.A.T., G.W., M.P., and J.K. acknowledge support from the Carbon Hub, a non-profit institute which receives corporate funding from Shell, Mitsubishi Corporation (Americas), and Prysmian. We thank the staff and facilities of the Shared Equipment Authority at Rice University, including the Electron Microscopy Center.

## Author contributions

N.K. and J.K. conceived the project. N.K. and Y.I. performed measurements, analyzed experimental data, and prepared the manuscript under the supervision of Y.Y., G.W., K.Y., and J.K. O.S.D. and L.W.T. prepared the fiber samples, and M.A.T. conducted doping, under the supervision of M.P. All authors discussed the results and commented on the manuscript.

## Competing interests

The authors declare the following competing financial interest(s): M.P. has a financial interest in DexMat, Inc.,which is commercializing solution-spun carbon nanotube fibers.
