## [Peer Review File · Nature Communications]

REVIEWER COMMENTS

Reviewer #1 (Remarks to the Author):

The authors spun the DWNTs into fiber and investigated its thermoelectric (TE) properties. They tuned Fermi energy (E_F) through chemical treatment to maximize Seebeck coefficient (S), obtaining the power factor (PF) as high as 14 mW/mK^2 . The authors developed a theoretical model explain this high PF value and validated it with finer E_F tuning using electrolyte gating. The Seebeck coefficient of the CNT fibers is in the range of $10\text{-}100 \text{ } \mu\text{V/K}$, which is not high compared with many other TE materials. The high value of PF results from ultrahigh electrical conductivity of the CNT fiber. Since the CNT fibers have ultrahigh thermal conductivity too, they are not the proper TE material for the application as TE generator because the figure-of-merit (ZT) is pretty low. As mentioned by the authors, the CNT fibers might be a good candidate for active cooling. Generally, the Peltier effect is related with ZT , but more sensitive to S , while not very sensitive to thermal conductivity. It is known that the maximum hot-side heat flow rate in active cooling is proportional to the effective thermal conductivity. Higher effective thermal conductivity requires large thermal conductivity and PF. The readers will be very interested to see how good the CNT fiber for thermoelectric cooling. The authors demonstrated that the CNT threads (bundle of CNT fibers) can be used as TE generator. Therefore, it should not be difficult for authors to demonstrate the device as TE cooler. It is suggested that the authors include the demonstration of CNT threads as TE cooler in this paper.

Other suggestions.

1. Page 3, line 41. "from the hot side to the cold site" should be "from the cold side to the hot site"
2. Page 8, line 142. give the definition of CNP
3. Page 12, Fig. 1e. change the title of x-axis from "Power factor" to "Power Factor"
4. Page 16, Fig. 3b. replace it by the optical image with higher quality
5. Page 19, line 298. give the length of the fiber
6. Page 20, line 337. "(Figure 2d)" should be "(Figure 3d)"
7. Page 21, line 339. "Figure 2d" is not right, is it "Figure S6b"?
8. Page 21, line 341. Add " Ω " to 300.

Reviewer #2 (Remarks to the Author):

Organic thermoelectric materials are of great interest in the fields of waste heat recovery and active cooling. Carbon nanotubes have unique mechanical and electrical properties, which have been widely studied for thermoelectric applications. The authors reported that the carbon nanotube fibers exhibited giant thermoelectric power factor, which was due to the one-dimensional quantum confinement of charge carriers (shifting the Fermi energy close to the van Hove singularity in the electronic density of states). The power factor obtained in this paper is impressive. But after a careful reading, I am not convinced by the argument approaching van Hove singularity leads to the high power factor. Major revisions are necessary before I make the decision to recommend its publication in the Nature communications.

1, The authors claim that the carbon nanotube fibers could be centimeter-long. However, it is hard to understand why the length of the carbon nanotube fiber need to be emphasized. Carbon nanotube fibers or yarns at meters or tens of meters scale have been often reported.

2, It is not reader friendly because of the missing of the preparation and characterization of the

carbon nanotube fibers. The synthesis and characterization of the carbon nanotube fibers should be discussed in the main text other than referring to other papers.

3, The authors claimed that their PF of $14 \text{ mWm}^{-1}\text{K}^{-2}$ is comparable to the highest values reported for 2D materials and of bulk materials in page 5 line 84-84. However, the maximum PF values at room temperature in the reference 10 and 11 are about $36 \text{ mWm}^{-1}\text{K}^{-2}$ and $26 \text{ mWm}^{-1}\text{K}^{-2}$, respectively. These values are all over two times higher than the PF of carbon nanotube fibers obtained in this paper. "comparable" may not be an accurate word to describe the large differences between them.

4, S1 and S2 were used to represent both the equations and the CNTs, which makes it hard to understand what the authors were trying to say in the supporting information.

5, The thermoelectric properties of DWCNT can be directly calculated with ATK. Why the authors use a simplified model to approximate the thermoelectric properties of DWCNT?

6, The theoretical calculation is almost independent of the experimental results. Where is the position of the Fermi level in the experimental samples? Does it locate the position as the calculation performed? If not, one cannot use the calculation results to explain the experimental results.

7, "weavability" should be changed with "weavability". Please check the typos.

Reviewer #3 (Remarks to the Author):

The manuscript "Centimeter-Long Weavable Fibers of Carbon Nanotubes with Giant Thermoelectric Power Factor" reported that the longitudinally centimeter-scaled carbon nanotubes with tuning Fermi level have a high electrical and thermal conductivity, generating a high power factor. I recommend addressing the following comments.

1. The author should have used various chirality carbon nanotubes, which means that their band gaps are different. It is unclear how to tune the Fermi levels of all the carbon nanotubes whose electronic band structures are dissimilar. Furthermore, it is necessary to state how to control electronic transport properties at the contact between carbon nanotubes.
2. One of the main reasons for the high power factor is the extremely high electrical conductivity. I recommend measuring it in another lab, if possible, to confirm the result.
3. The Seebeck coefficient is strongly dependent on the contact. For example, the contact may create Schottky barriers, which will affect the measurement results. It is recommended to use different contact materials in the measurements to see if the result is reliable.
4. It would be useful to provide more information such as SEM, XRD, and TEM images regarding the CNT fiber after the annealing processes.
5. It is necessary to provide more quantitative descriptions about the reasons for changes of the Seebeck coefficient and electrical conductivity of CNT fibers due to dedoping process at the annealing temperature of 350 K and 500 K.
6. It is necessary to provide more information regarding the thermoelectric properties of the thermoelectric generator (TEG). Make sure the flexibility by measuring the resistance and output performance of the TEG when they are bent. In addition, it is necessary to provide the charging time of the capacitor to turn on the LED bulb as well as the stability and the reproducibility of test results of the TEG.

Response to Reviewer 1's Comments

We are grateful to the reviewer for carefully reading our manuscript and making detailed comments. Below, we provide a point-by-point response to the comments:

Major Comment:

As mentioned by the authors, the CNT fibers might be a good candidate for active cooling. Generally, the Peltier effect is related with ZT, but more sensitive to S, while not very sensitive to thermal conductivity. It is known that the maximum hot-side heat flow rate in active cooling is proportional to the effective thermal conductivity. Higher effective thermal conductivity requires large thermal conductivity and PF. The readers will be very interested to see how good the CNT fiber for thermoelectric cooling. The authors demonstrated that the CNT threads (bundle of CNT fibers) can be used as TE generator. Therefore, it should not be difficult for authors to demonstrate the device as TE cooler. It is suggested that the authors include the demonstration of CNT threads as TE cooler in this paper.

Response to the major comment: We fully agree with the reviewer that a demonstration of active cooling using CNT fibers will be very interesting and a natural extension of the current work. However, such a study will require a substantial amount of work, worth a publication on its own. Furthermore, note that active cooling is not the main subject of the current paper; we mention it as an important future application of the findings we are reporting in this paper. Therefore, we conclude that a demonstration of active cooling is outside the scope of the current paper. Nevertheless, since we are already planning a follow-up study on active cooling, below we describe our motivation, approach, and required future work.

The study of active cooling by CNTs is largely unexplored despite the fact that carbon-based materials have been suggested as promising candidates for such applications¹. This motivates us to conduct a thorough investigation of active cooling by CNT fibers first before moving on to demonstrations using CNT threads. For example, in addition to Peltier measurements, it is important to understand and optimize the thermal conductivity of CNT fibers, as well as to measure the effective thermal conductivity as a function of temperature difference and electrical current. As shown in Page 3 line 44, the effective thermal conductivity consists of a passive part, which is the thermal conductivity, and an active part, which is proportional to the power factor. Therefore, it will be interesting to study the Fermi-energy dependence of the thermal conductivity as well as the effective thermal conductivity. Furthermore, we would like to optimize the device design with CNT threads for the active

cooling application because the main purpose of the device in this study was to demonstrate the weavability and scalability. Realizing the ultimate potential of a high-performance active cooling device will require detailed thermal engineering of the CNT fiber length and spacing, choice of electrical connection (or *n*-type) material with proper thermal and electrical properties, choice of thermal resistances of the heating and cooling reservoirs in the measurement, and analysis of parasitic heat losses, all of which are outside the scope of the current work.

Comment #1:

Page 3, line 41. “from the hot side to the cold site” should be “from the cold side to the hot site”

Response to comment #1: As the reviewer is well aware, in the conventional refrigeration mode, heat is pumped from the cold side to the hot side (see also Page 3 line 41-43). However, in the active cooling mode, the Peltier effect is used to enhance the heat flow rates from the hot side to the cold side, which is why having a high thermal conductivity becomes the advantage. Figure 1 of Ref. ² clarifies this point. However, this comment helped us realize that we did not make this point clear in the original manuscript. **We have thus changed the phrase “from the hot side to the cold side” to “from the hot side to the ambient temperature” (Page 3 line 40).**

Comment #2:

Page 8, line 142. give the definition of CNP

Response to comment #2: If the reviewer means to define the abbreviation, the definition is already given in Page 4 line 69. If the reviewer means to define the meaning of CNP, it is the middle of the band gap that separates the *p*-type and *n*-type regions, namely, the energy where the Fermi energy becomes zero. **In the revised manuscript, we have added “i.e., $E_F = 0$ eV” (Page 10 line 176) after “CNP” to clarify its definition.**

Comment #3:

Page 12, Fig. 1e. change the title of x-axis from “Power factor” to ‘Power Factor’

Response to comment #3: We thank the reviewer for pointing this out. It has been corrected.

Comment #4:

Page 16, Fig. 3b. replace it by the optical image with higher quality

Response to comment #4: We thank the reviewer for this suggestion. The picture has been replaced with a higher-quality image.

Comment #5:

Page 19, line 298. give the length of the fiber

Response to comment #5: We have added “The channel length was 0.6-1.2 cm.” to the **Methods section (Page 23 line 374)**. The channel length varied from device to device because the devices were manually made.

Comment #6:

Page 20, line 337. “(Figure 2d)” should be “(Figure 3d)”

Response to comment #6: We thank the reviewer for pointing out this error. We have changed it to Figure 3f, since we had to revise Figure 3 to address Reviewer #3’s comment #6.

Comment #7:

Page 21, line 339. “Figure 2d” is not right, is it “Figure S6b”?

Response to comment #7: We thank the reviewer for pointing out this error. The correct figure number is Figure 3f. This has been corrected.

Comment #8:

Page 21, line 341. Add “ Ω ” to 300.

Response to comment #8: It has been added.

Response to Reviewer 2's Comments

We are grateful to the reviewer for carefully reading our manuscript and making detailed comments. In the following, we respond to each of his/her comments in detail:

Major Comment:

Organic thermoelectric materials are of great interest in the fields of waste heat recovery and active cooling. Carbon nanotubes have unique mechanical and electrical properties, which have been widely studied for thermoelectric applications. The authors reported that the carbon nanotube fibers exhibited giant thermoelectric power factor, which was due to the one-dimensional quantum confinement of charge carriers (shifting the Fermi energy close to the van Hove singularity in the electronic density of states). The power factor obtained in this paper is impressive. But after a careful reading, I am not convinced by the argument approaching van Hove singularity leads to the high power factor.

Response to comment the major comment: We thank the reviewer for this insightful comment. We did not intend to argue that the high power factor solely arose from approaching a van Hove singularity in the electronic density of states, but we believe that having an ultrahigh electrical conductivity is also important. However, we agree with the reviewer that it was not clear in the original manuscript. **Therefore, we have modified the introduction and conclusion parts accordingly and added a new section entitled “origin of the giant power factor (Page 11, line 198 – Page 12, line 224).”** Below is the new section added:

Origin of the giant power factor. The observed giant PF can be explained as a combined effect of excellent sample morphology, which led to the ultrahigh σ , and the ability to tune E_F to the vicinity of a VHS. To compare the observed TE properties of our CNT fibers with those of previous CNT samples, Supplementary Figures 17a and b show the maximum values of S and PF , respectively, in each sample as a function of σ . The maximum value of S in this study is comparable to S values reported in previous studies (Supplementary Figure 17a), while the maximum PF value in this study is distinctly higher than other PF values (Supplementary Figure 17b), suggesting that the ultrahigh conductivity was key to the achieved giant PF . Although σ for individual metallic SWCNTs has been reported to be $\sim 100 \text{ MSm}^{-1}$ ^{35,36}, σ for CNT assemblies used in TE studies has been lower than 1 MSm^{-1} . In this study, the excellent morphology of the aligned CNT fibers led to the well-preserved σ of $>10 \text{ MSm}^{-1}$ even in a macroscopic assembly, resulting in the ultrahigh value of PF . PF values for CNT assemblies can be further enhanced in the future by improving the sample morphology, better preserving the σ of individual CNTs.

In addition to improving σ , tuning E_F is also crucial for maximizing PF . Supplementary Figures 17c and d show not only the maximum values but also other experimental values from E_F tuning measurements for this study and for samples from Ref. 22. As shown in Supplementary Figure 17d, within the same sample, increasing σ does not always result in improving PF ; in this study, increasing σ decreased both S and PF . Furthermore, the decrease or increase of S and PF with respect to σ is not monotonic; S as well as PF show a peak when E_F is near a VHS in the DOS, as demonstrated in Figure 2. Therefore, it is important to understand the E_F dependence of TE properties, and tune E_F accordingly to maximize PF .

These results highlight the promising properties of CNTs as a TE material. Conventionally, materials with high σ were considered undesirable for TE device applications because of the well-known trade-off between S and σ . However, as shown in this study, CNTs can provide relatively large S if E_F is tuned to the vicinity of a VHS in the DOS, in spite of having an ultrahigh σ . This is because S can be enhanced through a narrow carrier distribution¹³, which is achieved by 1D quantum confinement of charge carriers in the case of CNTs.

Supplementary Figure 17. Comparison of reported (a) Seebeck coefficient and (b) power factor for various CNT samples as a function of electrical conductivity. (b) Figure 1e without non-CNT references. Values are summarized in Supplementary Table 4. (c) Seebeck coefficient and (d) power factor with varying electrical conductivity for (6,5) SWCNTs, metal enriched SWCNTs and aligned metallic SWCNTs.

Comment #1:

The authors claim that the carbon nanotube fibers could be centimeter-long. However, it is hard to understand why the length of the carbon nanotube fiber need to be emphasized.

Carbon nanotube fibers or yarns at meters or tens of meters scale have been often reported.

Response to comment #1: The reviewer is absolutely correct that CNT fibers and yarns with meters and tens of meters of length is typical. The CNT fiber production method used in this study can produce hundreds of meters of continuous fiber in our lab at Rice and considerably longer fibers in a startup from Rice (DexMat). In an earlier version of our manuscript, we had used the adjective “large-scale” to describe these fibers, but a few co-authors felt that such a term was too generic. We thus switched to “centimeter-scale” because we used pieces of fibers that were a few centimeters long in the current thermoelectric study, which should be contrasted to earlier reports on high PF values that used microscale materials. In fact, 2D materials used in prior thermoelectric studies such as graphene, which achieved the highest reported PF at room temperature, were only micrometer-scale, not desirable for practical large-scale applications. On the other hand, we observed a giant PF in a truly macroscopic sample, and thus, we wanted to emphasize it as one of the strengths of our sample. From a materials production viewpoint, continuous fibers were made with lengths >100 m and then cut into centimeter-scale pieces for the TE measurements and thermoelectric generator demonstration. However, we now realize that readers with a background in material production might get confused, and we thank the reviewer for this comment. **Therefore, we have changed the expression “centimeter-scale” to “macroscopic” in the title, and added a few sentences to clarify this point in Page 8, line 129-135, as shown below.**

Note that the giant PF in this work was observed in macroscopic samples (~1 cm length), whereas the highest PF reported for 2D materials has been measured in microscopic samples (typically with dimensions of a few μm). CNT fibers were produced in continuous runs (over 100 m in total length, as discussed above) and cut into centimeter-scale pieces to conduct measurements. We further demonstrate this strength, i.e., scalability, in a later section by using CNT threads that were produced by plying multiple fibers via a continuous plying machine.

We have also added Supplementary Table 1, where we summarize dimensions of raw CNT materials, fibers, and threads, in order to avoid any confusion:

Supplementary Table 1. Dimensions of CNTs, CNT fibers, and CNT threads used in this study.

		CNTs	CNT fibers	CNT threads
For thermoelectric study	Production source	Meijo NC	Solution spinning CNTs	
	Average diameter	1.8 nm	8.9 μm	
	Length	12 μm (average)	> 100 m	
For TEG demonstration	Production source	Meijo NC		Plying 21 CNT fibers together
	Average diameter	1.8 nm		190 μm
	Length	7.4 μm (average)		> 100 m

Comment #2:

It is not reader friendly because of the missing of the preparation and characterization of the carbon nanotube fibers. The synthesis and characterization of the carbon nanotube fibers should be discussed in the main text other than referring to other papers.

Response to comment #2: We have added the following paragraph in Page 6, line 89-107, describing the method and basic characterization (TEM for raw CNTs, and SEM and Raman for produced CNT fibers):

CNTs used to fabricate fibers mainly contained double-wall CNTs (DWCNTs) with an average outer (inner) wall diameter of 1.8 ± 0.2 nm (0.9 ± 0.1 nm), measured by high resolution transmission electron microscopy (HR-TEM) (Supplementary Figure 1). The viscosity-averaged aspect ratio was $6.7 (\pm 0.1) \times 10^3$, measured using a capillary thinning extensional rheometer²⁴. A solution spinning method^{15,16} was used to spin CNTs into a continuous fiber. CNTs were first dissolved in chlorosulfonic acid (CSA) to create a spin dope²⁵. The dope was then filtered and extruded into a coagulant. Finally, the coagulated fiber was collected onto a rotating drum. This method produced meters (>100 m) of fiber with densely packed and highly aligned CNTs, as shown in Figure 1a. The average diameter of the fiber was determined to be 8.9 ± 0.9 μm by scanning electron microscopy (SEM). Dimensions of CNTs and the produced CNT fiber are summarized in Supplementary Table 1. Raman spectroscopy was performed on the produced CNT fibers (Supplementary Figures 2b and c). The average G-to-D ratio was 50 (with 532 nm excitation), demonstrating a low density of defects (Supplementary Figure 2b). The radial breathing mode (RBM) region of the Raman spectra indicates the presence of CNTs with diameters ranging from 0.8 to 2.0 nm (Supplementary Figure 2c), consistent with results from HR-TEM measurements. The CNTs inside the solution spun fiber have a high aspect ratio as well as a low impurity density, and are highly crystalline, leading to exceptional mechanical (a tensile strength of 4.2 ± 0.2 GPa) and electrical ($\sigma > 10$ MSm⁻¹) properties while retaining flexibility^{15,17}.

Comment #3:

The authors claimed that their PF of 14 mWm-1K-2 is comparable to the highest values reported for 2D materials and of bulk materials in page 5 line 84-84. However, the maximum PF values at room temperature in the reference 10 and 11 are about 36 mWm-1K-2 and 26 mWm-1K-2, respectively. These values are all over two times higher than the PF of carbon nanotube fibers obtained in this paper. “comparable” may not be an accurate word to describe the large differences between them.

Response to comment #3: We have changed the adjective “comparable” to “approaching” (Page 5 line 83 and Page 8 line 128).

Comment #4:

S1 and S2 were used to represent both the equations and the CNTs, which makes it hard to understand what the authors were trying to say in the supporting information.

Response to comment #4: We have changed S1 and S2, which were used for CNTs, to “SC1” and “SC2”, respectively.

Comment #5:

The thermoelectric properties of DWCNT can be directly calculated with ATK. Why the authors use a simplified model to approximate the thermoelectric properties of DWCNT?

Response to comment #5: DWCNTs generally have an incommensurate crystal structure (except rare cases), which leads to a quasi-periodic potential energy landscape without translational symmetry. As a result, it is not feasible to rigorously calculate properties such as their electronic structure, electrical conductance, and Seebeck coefficient with the atomistic simulation software we use (Quantum ATK, Synopsys Co.). It is thus inevitable to introduce approximations. On the other hand, the thermoelectric properties of SWCNTs using a tight-binding model with ATK is well established³.

Furthermore, interlayer interactions of DWCNTs are known to mainly result in dielectric screening of Coulomb interactions, which is not strong enough to result in the formation of new energy bands⁴. This can be experimentally confirmed by Supplementary Figure 7a, where the expected peaks of independent SWCNTs with corresponding inner and outer diameters explain the absorbance spectra for the annealed samples. Please note that the peaks in Supplementary Figure 7a are slightly red-shifted from the expected peaks due to dielectric screening.

Comment #6:

The theoretical calculation is almost independent of the experimental results. Where is the position of the Fermi level in the experimental samples? Does it locate the position as the calculation performed? If not, one cannot use the calculation results to explain the experimental results.

Response to comment #6: Yes, the position of the Fermi energy (E_F) in the experimental samples are located within the range where the calculation was performed. As stated in Page 8 line 147 of the original manuscript, we conducted systematic optical spectroscopy measurements to estimate the E_F of four chemically treated samples so that we can compare them with the calculations. Figures 2a, b, and c replot experimentally observed values as a

function of the estimated E_F on the right axis, indicating that the position of E_F in the experimental samples are located within the calculated range.

This comment helped us realize that we did not make this point clear in the original manuscript, since details are only discussed in the Supplementary Information. **Hence, we have added the following paragraph (Page 8 line 145 – Page 9 159)**, where we describe how we estimated the location of E_F in the samples:

We first estimated the E_F of four chemically treated fibers discussed in Figure 1 by conducting systematic optical spectroscopy measurements. Thin films of aligned CNTs produced by a facile blade coating technique²⁹ were used, because their size and absorption were more suited for optical measurements. This method also starts from dissolving CNTs in CSA as in the solution spinning method, and then produce films instead of fibers. We used the same raw CNT materials and ensured that the chemical treatments (e.g., CSA concentration, doping, and annealing conditions) were identical to the treatments to the fibers discussed above. Supplementary Figure 7a compares expected absorbance peak positions based on the diameter distribution^{30,31} determined by the TEM analysis with absorbance spectra for the four films annealed at 500 °C, annealed at 350 °C, as-produced, and ICl doped, respectively. The E_F of the annealed (as-produced and doped) samples was estimated to be in the vicinity of the first VHS of the outer (inner) semiconducting tubes based on absorbance peak analysis (Supplementary Figure 7b). More details are discussed in Supplementary Note 1. σ , S , and PF measured in Figure 1 are plotted as a function of the estimated E_F in Figures 2a, b, and c, respectively.

Comment #7:

“weavability” should be changed with “weavability”. Please check the typos.

Response to comment #7: We thank the reviewer for pointing out this typo. It has been corrected.

Response to Reviewer 3's Comments

We are grateful to the reviewer for carefully reading our manuscript and making detailed comments. In the following, we respond to each of his/her comments in detail:

Comment #1:

1. The author should have used various chirality carbon nanotubes, which means that their band gaps are different. It is unclear how to tune the Fermi levels of all the carbon nanotubes whose electronic band structures are dissimilar.
2. Furthermore, it is necessary to state how to control electronic transport properties at the contact between carbon nanotubes.

Response to comment #1:

1. We are not entirely sure whether we have correctly understood this question, but the Fermi energy (E_F) can be tuned simultaneously for all chirality nanotubes present in the fiber. For example, if a chemical treatment shifts E_F by 0.2 eV, all nanotubes in the fiber experience the same 0.2 eV shift equally. Now, whether this 0.2 eV shift is large enough to move E_F into the valence or conduction band depends on the bandgap, which is determined by the diameter⁵.

The DWCNTs used in this study were not chirality-sorted, but they had a diameter distribution; the average outer- (inner-) wall diameter was 1.8 ± 0.2 nm (0.9 ± 0.1 nm), determined by high resolution TEM analysis. The inner-wall diameter distribution was confirmed by an absorbance spectrum for the raw CNT material (Supplementary Figure 6), and the outer-wall diameter distribution is shown in Figure 1c of Ref. ⁶, where the same raw CNT material was used. In short, a diameter histogram for our sample would show two distinct peaks, at around 0.9 nm and 1.8 nm, instead of a single-peak distribution typical of a SWCNT ensemble. Because the bandgap energy of a CNT is determined by the diameter⁵, a narrow distribution in diameter results in a narrow distribution in bandgap. This assumption was further confirmed by the optical study (Supplementary Figure 7a), where the absorbance spectra can be interpreted as a superposition of spectra for semiconducting and metallic SWCNTs for outer wall and inner wall.

2. Electronic transport in assemblies of CNTs occurs through two distinct mechanisms: intertube transport (transport between CNTs) and intratube transport (for which the conductivity is usually limited by electron-phonon scattering at high temperatures)⁷.

One way to determine which of these two mechanisms is dominant under given conditions is via a study of the temperature (T) dependence of σ . “Semiconducting” behavior (σ rising with increasing T) can be attributed to intertube transport, whereas “metallic” behavior (σ decreasing with increasing T) can be attributed to intratube transport. In typical macroscopic CNT samples, the T dependence of σ shows semiconducting behavior up to room temperature⁸. However, as shown in Figure 4a of Ref. ⁷, the excellent sample morphology of our CNT fibers (a high degree of CNT alignment, a high density, a high CNT aspect ratio, and a low density of impurities) led to the metallic behavior above 200 K. Therefore, at room temperature (300 K), where our measurements were conducted, our data is not significantly affected by intertube transport.

Comment #2:

One of the main reasons for the high power factor is the extremely high electrical conductivity. I recommend measuring it in another lab, if possible, to confirm the result.

Response to comment #2: We measured the electrical conductivity of the as-produced fiber (reported as 11 ± 2 MS/m in the manuscript) in three different labs to confirm the reproducibility. The results are summarized in Table R1. The as-produced continuous fiber (> 100 m) was cut into pieces and distributed among three labs. All of them used a four-probe method so that the contact resistance was negligible, and the measurements were conducted at room temperature. As shown in Table R1, all three labs confirmed the reported value in the manuscript.

Table R1. Electrical conductivity of the as-produced fiber measured by different labs.

Measured by	Yanagi lab (Tokyo Metropolitan University)		Kono lab (Rice University)	Pasquali lab (Rice University)
	Reported in paper	Re-measurement in April 2021		
Method	Four probe measurement			
Temperature	Room temperature			
Pressure	Vacuum ($\sim 10^{-3}$ Pa)		Vacuum ($\sim 10^{-3}$ Torr)	In air, 1 atm
Channel length	8.8 mm	8.7-9.0 mm	3.7 mm	7.0 cm
Conductivity (MS/m)	11 ± 2	11 ± 2	10 ± 1	11 ± 2

Comment #3:

The Seebeck coefficient is strongly dependent on the contact. For example, the contact may create Schottky barriers, which will affect the measurement results. It is recommended to use different contact materials in the measurements to see if the result is reliable.

Response to comment #3: We thank the reviewer for this thoughtful recommendation. In order to address this concern, we have conducted additional measurements as follows. We measured the Seebeck coefficient of the as-produced CNT fiber using different contact materials. This CNT fiber was identical to the “as produced” sample in the paper. Here we checked the Seebeck coefficients of the fiber using three different contacts: silver paste, carbon paste, and gold paste. The device structure and the measurement method were the same as described in the Methods section. The results are summarized in Table R2. As shown in Table R2, the Seebeck coefficients were the same within the experimental errors, and thus, we conclude that any contact effect is negligible in this study.

In the revised manuscript, we have added the sentence “We used three different contact materials, silver (D-500 DOTITE, Fujikura Kasei Co.), carbon (DOTITE XC-12, Nisshin EM Co.), and gold (No. 8560, Tokuriki Honten Co., Ltd.), to ensure the independence of the measured Seebeck coefficient values of the contact material.” to the Methods section (Page 23, line 381-383).

Table R2. Seebeck coefficient of the as-produced CNT fiber measured using different contact materials

Contact			Seebeck coefficient ($\mu\text{V/K}$)
Material	Product details	Resistivity (Ohm cm)	
Silver	D-500 DOTITE (Fujikura Kasei Co.)	8×10^{-5}	23 ± 1
Carbon	DOTITE XC-12 (Nisshin EM Co.)	1×10^{-2}	23 ± 1
Gold	No. 8560 (Tokuriki Honten Co., Ltd.)	5×10^{-3}	23 ± 1

Comment #4:

It would be useful to provide more information such as SEM, XRD, and TEM images regarding the CNT fiber after the annealing processes.

Response to comment #4: We thank the reviewer for this suggestion. To address this question, we have added SEM and Raman characterization data for the CNT fibers after the annealing process to the Supplementary Information (Supplementary Figure 3).

CNT fibers that we produce using the solution spinning method typically have diameters of 10 μm and cannot be imaged via TEM. Therefore, we use SEM to characterize diameter and surface morphology. We conduct TEM only on raw CNT materials, before spinning them into fibers, which can be seen in Supplementary Figure 1.

This comment helped us realize that we did not make clear distinctions between the characterization of the raw CNT materials and the CNT fiber. **We have rearranged Supplementary Figures so that Supplementary Figure 1 shows the characterization of the raw CNT materials, Supplementary Figure 2 is for the as-produced CNT fibers, and Supplementary Figure 3 is for the annealed CNT fibers. Further, we have added Supplementary Table 1 to summarize dimensions for the raw CNT material, the spun CNT fiber, and the CNT threads, in order to avoid any confusion between them.** Supplementary Figures 1, 2 and 3 and Supplementary Table 1 are reproduced below:

Supplementary Figure 1. High resolution transmission electron microscopy image of the raw CNT material. The average outer (inner) wall diameter was determined to be 1.8 ± 0.2 nm (0.9 ± 0.1 nm) and the average number of walls was 1.9.

Supplementary Figure 2. Characterization of as-produced CNT fibers. (a) SEM image of the surface morphology of the fiber produced by solution spinning method, depicting highly aligned CNT bundles. The average diameter was measured to be 8.9 ± 0.9 μm . (b) Raman spectra for the as-produced CNT fibers excited by 532 nm (green), 633 nm (yellow), and 785 nm (blue) lasers. The average G-to-D ratio was determined to be 50, 45 and 25 for the 532 nm, 633 nm and 785 nm excitations. (c) Raman spectra in the radial breathing mode region. Raman shift was converted into the diameter using the relationship proposed in Ref. ⁹.

Supplementary Figure 3. Characterization of CNT fibers annealed at (a) & (b) 350 °C and (c) & (d) 500 °C. (a) SEM image of the fiber annealed at 350 °C. (b) Raman spectra for the fiber annealed at 350 °C excited by 532 nm (green), 633 nm (yellow), and 785 nm (blue) lasers. The average G-to-D ratio was determined to be 24, 49, and 40 for the 532 nm, 633 nm, and 785 nm excitation wavelengths, respectively. The decrease in G-to-D ratio compared to the as-produced fibers is mainly attributed to shoulders of the G-peak giving higher background, not to induced defects. (c) SEM image of the fiber annealed at 500 °C. (d) Raman spectra for the fibers annealed at 500 °C. The average G-to-D ratio was determined to be 20, 48, and 29 for the 532 nm, 633 nm, and 785 nm excitation wavelengths, respectively.

Supplementary Table 1. Dimensions of CNTs, CNT fibers, and CNT threads used in this study.

		CNTs	CNT fibers	CNT threads
For thermoelectric study	Production source	Meijo NC	Solution spinning CNTs	
	Average diameter	1.8 nm	8.9 μm	
	Length	12 μm (average)	> 100 m	
For TEG demonstration	Production source	Meijo NC		Plying 21 CNT filaments together

	Average diameter	1.8 nm		190 μm
	Length	7.4 μm (average)		> 100 m

Comment #5:

It is necessary to provide more quantitative descriptions about the reasons for changes of the Seebeck coefficient and electrical conductivity of CNT fibers due to dedoping process at the annealing temperature of 350 K and 500 K.

Response to comment #5: The as-produced fiber from the solution spinning method is heavily doped by chlorosulfonic acid. Annealing the as-produced fiber desorbs the chlorosulfonic acid with an activation energy that is proportional to $k_{\text{B}}T$, where k_{B} is the Boltzmann constant and T is the temperature¹⁰. Therefore, annealing at a higher temperature leads to desorbing more dopant, leading to stronger dedoping (moving the Fermi energy closer to the charge neutrality point). We chose 350 °C as the first annealing temperature to ensure the annealing temperature is well above the boiling point of chlorosulfonic acid (~152 °C).

Quantitative determination of the Fermi energy shift difference between the sample annealed at 350 °C and the one annealed at 500 °C is more challenging. In this study, it was limited to a quantitative estimation based on the optical absorption measurements (Supplementary Figure 7). Furthermore, because the electrical conductivity monotonically increases with doping in carbon nanotubes, the decrease of the electrical conductivity after annealing is usually explained as a dedoping effect.

Another way to estimate the Fermi energy shift is to use Raman spectroscopy¹¹. Figure R1 shows Raman spectra for the fibers annealed at 350 °C (top) and 500 °C (bottom). The G-band consists of four peaks: the upper (G^+) and lower (G^-) branches of the inner (G_{i}) and outer (G_{o}) carbon nanotubes¹¹. The shift of the upper branch of the outer carbon nanotubes (G_{o}^+) is related to the Fermi energy shift; G_{o}^+ shows a blue-shift with doping. The sample annealed at 500 °C showed a red-shift compared to the sample annealed at 350 °C, consistent with the expected dedoping effect. Note that we do not expect a large Fermi energy shift between the two samples based on the optical study (Supplementary Figure 7), which was consistent with the small red-shift observed in the Raman spectra.

Figure R1. Raman spectra for the fiber annealed at 350 °C (top) and the fiber annealed at 500 °C (bottom). Lorentzians were used to fit the upper (G^+) and lower (G^-) branches of the inner (G_i) and outer (G_o) carbon nanotubes¹¹. The peak position of the upper branch of outer carbon nanotubes (G_o^+) and inner carbon nanotubes (G_i^+) are indicated in units of cm^{-1} .

Comment #6:

It is necessary to provide more information regarding the thermoelectric properties of the thermoelectric generator (TEG). Make sure the flexibility by measuring the resistance and output performance of the TEG when they are bent. In addition, it is necessary to provide the charging time of the capacitor to turn on the LED bulb as well as the stability and the reproducibility of test results of the TEG.

Response to comment #6: We thank the reviewer for these insightful suggestions. **We have conducted a series of additional measurements** to address the reviewer's points:

1. **Flexibility.** In order to ensure the performance of the thermoelectric generator (TEG) device while they are bent, we bent one TEG unit (consisting of fifteen CNT threads) with a bending radius of 3.18 mm (see Supplementary **Error! Reference source not found.c** and d), and applied a temperature difference while maintaining the bending. Supplementary **Error! Reference source not found.e** shows a generated voltage as a function of the temperature difference without bending (black) and with bending (red). The slope of the two fitting curves differs only by 3.8 %, verifying that there is essentially no degradation in device performance due to bending. This information has been added to the Supplementary Information.

Supplementary Figure 18. (a) No bending condition, where one TEG unit (consisting of fifteen CNT threads) was suspended between a hotplate and a heat sink. The temperature difference was created by increasing the hot side temperature (T_{hot}) by the hotplate, while no modification was added to the cold side (T_{cold}). (b) Schematic image of (a). (c) Bending condition, where the device was bent with a bending radius of 3.18 mm. The hot side was directly underneath the inserted bar. This position (with the bending radius of 3.18 mm) was fixed throughout the temperature dependence of generated voltage measurement. (d) Schematic image of (c). (e) Generated voltage from one TEG unit as a function of the temperature difference $T_{hot} - T_{cold}$ with no bending (black) and with bending (red). The fitting gives the total Seebeck coefficient for one TEG unit of 0.26 ± 0.01 mV/K for no bending and that of 0.25 ± 0.01 mV/K with bending.

- Flexibility and Stability.** The electrical conductivity of the TEG unit was measured with varying bending radii. Figure R2a shows the bending radius dependence of the conductivity, normalized by that of the unbent device. The conductivity stayed within $\pm 2\%$ of that of the unbent device up to a bending radius of 0.1 mm, demonstrating excellent flexibility. Figure R2b shows the normalized conductivity as a function of the number of bending events; the bending radius was 0.1 mm. The conductivity stayed within $\pm 1.9\%$ of that of the original state for 200 times of bending, demonstrating great stability. This information has been added to the main text.

Figure R2. (a) Electrical conductivity σ of one TEG unit as a function of bending radius, normalized by the conductivity without bending σ_0 . The inset shows a schematic and a picture of the device during the measurement. (b) σ as a function of bending number, normalized by that of the original state σ_0 . The bending radius was 0.1 mm. The inset shows the schematic or the measurement procedure.

- 3. Reproducibility.** The demonstration of lighting an LED by four TEG units was reproduced multiple times. We will attach three videos of the demonstration filmed on 09/23/2020, 10/05/2020, and 05/11/2021 (entitled as “IMG_1250”, “IMG_1364”, and “IMG_2371”, respectively). We did not edit the videos so that the reviewer can check the original data of them easily, thus so please do not bother background voices. Figures R3a, b, and c show the LED that was driven by four TEG units on 09/23/2020, 10/05/2020, and 05/11/2021, respectively. Note that the TEG successfully turned on the LED after more than half year on 05/11/2021, demonstrating indisputable reproducibility and stability.

Figure R3. Screenshots of videos of the demonstration filmed on (a) 09/23/2020, (b) 10/05/2020, and (c) 05/11/2021 (entitled as “IMG_1250”, “IMG_1364”, and “IMG_2371”, respectively), where LED

was turned on. (a) Screenshots were taken at 0:00:08 LED off (left) and on (right). (b) Screenshots were taken at 0:00:08. LED off (top) and on (bottom). The picture also shows capacitors, next to the LED. (c) Screenshots were taken at 0:00:24. LED off (left) and on (right). The picture also shows the capacitors and a multimeter.

4. **Charging time.** In an RC charging circuit, where a capacitor gets charged by a voltage source through the resistance, the voltage across the capacitor (V_c) at any instant in time during the charging period is given as

$$V_c = V_s \left(1 - \exp\left(-\frac{t}{\tau}\right) \right),$$

where V_s is the source voltage, t is the time, and τ is the time constant, defined as $\tau = RC$, where R is the resistance and C is the capacitance. This equation indicates that V_c is 98.2 % charged when $t = 4\tau$. In our case, the time constant is 4.95 seconds, and the charging time depends on V_s .

Furthermore, we have revised Figure 3 and added a paragraph in the main text describing the flexibility test. The new paragraph and the revised Figure 3 are reproduced below:

We investigated the flexibility of this device by performing bending tests. The device was wound around cylinders with specific diameters, as shown in the inset of Figure 3c, and the electrical conductivity (σ) at each bending radius was compared to that of the original state without bending (σ_0). Figure 3c shows that no significant change (less than 2 %) occurred up to a bending radius of 0.1 mm. We repeated the bending for 200 times with a fixed bending radius of 0.1 mm. As shown in Figure 3d, the conductivity did not change more than 1.9%. Furthermore, we applied a temperature difference to the device while it was bent. Supplementary Figure 18e shows a generated voltage as a function of the temperature difference during bending, verifying that there was essentially no degradation in device performance due to bending.

Revised Figure 3. Textile TE generator based on CNT threads sewn into a fabric. (a) Schematic and (b) photograph of the device. CNT threads (*p*-type thermoelectric generator) and steel threads (for electrical connection) were sewn into a fabric using a sewing machine. The CNT threads were connected electrically in series and thermally in parallel. A temperature difference ΔT was applied by heating one side (T_{hot}) while keeping the other side (T_{cold}) at room temperature. (c) Electrical conductivity σ of one TEG unit as a function of bending radius, normalized by the conductivity without bending σ_0 . The inset shows a schematic and a picture of the device during the measurement. (d) σ as a function of bending number, normalized by that of the original state (σ_0). The bending radius was 0.8 mm. The inset shows a schematic of the measurement procedure. (e) The entire device was connected to an LED through an amplification circuit and a capacitor. The LED turned on with a ΔT of ~ 50 K. (f) Output voltage and power as a function of applied ΔT . The inset shows a folded device, demonstrating its flexibility.

References used in responses

1. Zebarjadi, M. Electronic cooling using thermoelectric devices. *Appl. Phys. Lett.* **106**, 203506 (2015).
2. Adams, M. J., Verosky, M., Zebarjadi, M. & Heremans, J. P. Active Peltier Coolers Based on Correlated and Magnon-Drag Metals. *Phys. Rev. Appl.* **11**, 054008 (2019).
3. Ichinose, Y. *et al.* Solving the Thermoelectric Trade-Off Problem with Metallic Carbon Nanotubes. *Nano Lett.* **19**, 7370–7376 (2019).
4. Zhao, S. *et al.* Rayleigh scattering studies on inter-layer interactions in structure-defined individual double-wall carbon nanotubes. *Nano Res.* **7**, 1548–1555 (2014).
5. Weisman, B. R. & Kono, J. *Optical Properties of Carbon Nanotubes: A Volume Dedicated to the Memory of Professor Mildred Dresselhaus.* (World Scientific Publishing, 2019).
6. Taylor, L. W. *et al.* Improved properties, increased production, and the path to broad adoption of carbon nanotube fibers. *Carbon N. Y.* **171**, 689–694 (2021).
7. Behabtu, N. *et al.* Strong, Light, Multifunctional Fibers of Carbon Nanotubes with Ultrahigh Conductivity. *Science (80-.).* **339**, 182–186 (2013).
8. Dini, Y., Faure-Vincent, J. & Dijon, J. A unified electrical model based on experimental data to describe electrical transport in carbon nanotube-based materials. *Nano Res.* **13**, 1764–1779 (2020).
9. Maultzsch, J., Telg, H., Reich, S. & Thomsen, C. Radial breathing mode of single-walled carbon nanotubes: Optical transition energies and chiral-index assignment. *Phys. Rev. B - Condens. Matter Mater. Phys.* **72**, 205438 (2005).
10. Zhou, W. *et al.* Single wall carbon nanotube fibers extruded from super-acid suspensions: Preferred orientation, electrical, and thermal transport. *J. Appl. Phys.* **95**, 649–655 (2004).
11. Tristant, D. *et al.* Enlightening the ultrahigh electrical conductivities of doped double-wall carbon nanotube fibers by Raman spectroscopy and first-principles calculations. *Nanoscale* **8**, 19668–19676 (2016).

References used in transcribed sections from the revised main text

(reference numbers are identical to the ones in the revised main text)

13. Mahan, G. D. & Sofo, J. O. The best thermoelectric. *Proc. Natl. Acad. Sci.* **93**, 7436–7439 (1996).
15. Taylor, L. W. *et al.* Improved properties, increased production, and the path to broad adoption of carbon nanotube fibers. *Carbon* **171**, 689–694 (2021).
16. Behabtu, N. *et al.* Strong, light, multifunctional fibers of carbon nanotubes with

- ultrahigh conductivity. *Science* **339**, 182–186 (2013).
17. Adnan, M. *et al.* Bending behavior of cnt fibers and their scaling laws. *Soft Matter* **14**, 8284–8292 (2018).
 22. Ichinose, Y. *et al.* Solving the Thermoelectric Trade-Off Problem with Metallic Carbon Nanotubes. *Nano Lett.* **19**, 7370–7376 (2019).
 24. Tsentalovich, D. E. *et al.* Relationship of Extensional Viscosity and Liquid Crystalline Transition to Length Distribution in Carbon Nanotube Solutions. *Macromolecules* **49**, 681–689 (2016).
 25. Davis, V. A. *et al.* True solutions of single-walled carbon nanotubes for assembly into macroscopic materials. *Nat. Nanotechnol.* **4**, 830–834 (2009).
 29. Headrick, R. J. *et al.* Structure–property relations in carbon nanotube fibers by downscaling solution processing. *Adv. Mater.* **30**, 1704482 (2018).
 30. Weisman, R. B. & Bachilo, S.M. Dependence of optical transition energies on structure for single-walled carbon nanotubes in aqueous suspension: An empirical kataura plot. *Nano Lett.* **3**, 1235–1238 (2003).
 31. Weisman, R. B. & Kono, J. (eds.) *Optical Properties of Carbon Nanotubes: A Volume Dedicated to the Memory of Professor Mildred Dresselhaus* (World Scientific, Singapore, 2019).
 35. McEuen, P. L. & Park, J.-Y. Electron transport in single-walled carbon nanotubes. *MRS Bull.* **29**, 272–275 (2004).
 36. Park, J.-Y. *et al.* Electron- phonon scattering in metallic single-walled carbon nanotubes. *Nano Lett.* **4**, 517–520 (2004).

REVIEWER COMMENTS

Reviewer #1 (Remarks to the Author):

The authors have addressed my comments in the revisions. Look forward to seeing the demonstration of active cooling using CNT fibers.

Reviewer #2 (Remarks to the Author):

The authors have addressed all the questions well. The reviewer would like to recommend publishing this manuscript.

Reviewer #3 (Remarks to the Author):

The authors have adequately addressed most of my comments and improved the manuscript. It is recommended to address the following additional comments.

1. The Seebeck coefficient is pretty large despite the extremely high electrical conductivity. The van Hove singularity of CNTs whose chirality is different are dissimilar so it is not easy to improve the Seebeck coefficient by changing the doping level. It is challenging to uniformly control the doping level of the different chirality CNTs, which would influence the Seebeck and electrical conductivity. It is necessary to address these issues.
2. ZT needs to be plotted in the main manuscript so that readers can have the information without additional calculation.
3. LED has been turned on with 83 mV but I wonder if there is a LED that can be operated with such a low voltage. It is necessary to describe the specification of the LED. Also the LED was turned on for a very short period of time while TE is supposed to generate electricity continuously. This would be possible presumably with the circuitry. More detailed information is necessary for readers to have a better understanding about this work.

Response to Reviewer 3's Comments

We are grateful to the reviewer for carefully reading our revised manuscript and making valuable comments. In the following, we respond to each of his/her comments in detail:

Comment #1:

The Seebeck coefficient is pretty large despite the extremely high electrical conductivity. The van Hove singularity of CNTs whose chirality is different are dissimilar so it is not easy to improve the Seebeck coefficient by changing the doping level. It is challenging to uniformly control the doping level of the different chirality CNTs, which would influence the Seebeck and electrical conductivity. It is necessary to address these issues.

Response to comment #1:

We thank the reviewer for making this helpful comment. This comment actually deals with two separate issues, which we individually address below:

(A) *“The van Hove singularity of CNTs whose chirality is different are dissimilar so it is not easy to improve the Seebeck coefficient by changing the doping level.”* The reviewer is correct that, since different (n,m) species have VHSs at different energies, it is not possible to simultaneously optimize the doping level for different chiralities. However, this is true *only if* we treat different chirality tubes separately. For example, as shown in Figure R1a, the Seebeck coefficient (S_{ind}) becomes maximum for (22,0) when $E_F = -0.05$ eV, while (13,13)'s maximum occurs at $E_F = -0.57$ eV. However, we can model our sample as an assembly of nanotubes that are electrically connected in parallel. In this combined system, as shown in Figure R1b, S_{tot} becomes maximum when $E_F = -0.16$ eV, which is distinct from both -0.05 eV and -0.57 eV. We emphasize that our model is based on what has previously been used successfully for simulating CNT samples containing multiple chiralities.¹⁻³

This comment helped us realize that we did not clarify the difference between the combined model and models based on individual chiralities in our previous manuscript.

Hence, we have added subscripts “ind” to thermoelectric quantities (e.g., S_{ind}) obtained from the models based on individual chiralities, and subscripts “tot” (e.g., S_{tot}) to quantities obtained from the combined model. Furthermore, we have modified the main text (Page 12, line 226-227) to avoid any confusion.

However, as shown in this study, CNTs can provide relatively large S if E_F is properly tuned, in spite of having an ultrahigh σ .

Figure R1. (a) Calculated Seebeck coefficient S_{ind} for (22,0) (red solid line) and (13,13) (gray solid line) as a function of Fermi energy E_F . S_{ind} for (22,0) was multiplied by 0.2 for clarity. Data was extracted from Supplementary Figure 8. See Supplementary Note 2 for calculation details. (b) S_{tot} based on the combined model, calculated using Supplementary Equation (11) for (22,0) and (13,13).

(B) “It is challenging to uniformly control the doping level of the different chirality CNTs”. First, while we understand the reviewer’s concern, we point out the important fact that the work function of CNTs is essentially independent of the chirality.⁴⁻⁶ Therefore, the relationship between the E_F shift and the doping level is common to all chiralities. Second, we experimentally assessed the E_F position by monitoring the suppression of optical absorption peaks caused by Pauli blocking, as summarized in Figure R2. Our TEM analysis showed two distinct diameter peaks, at around 0.9 nm and 1.8 nm, corresponding to the inner and outer tubes. Then the empirical Kataura plot⁷ allowed us to identify different optical transition energies. Figure R2a shows an example for the outer-wall diameter, where a red shaded area indicates the diameter range from TEM. Note that we did not observe the E_{11}^S peak in the as-produced film (Figure R2d) due to Pauli blocking; doping shifted E_F into the valence band (Figure R2e). See Supplementary Note 1 for more details.

It is important to point out that Figure R2b (top) involves many chiralities: it contains 28 chiralities ranging from (15,8) (1.606 nm in diameter) to (15,14) (1.994 nm). Therefore, the peak in Figure R2b (bottom) is inhomogeneously broadened due to the presence of multiple chiralities. However, there is no visible peak in Figure R2d (bottom), suggesting that E_F was effectively shifted into the valence band by doping for all semiconducting chiralities within this diameter range. Therefore, although we agree with the reviewer that there would in principle be small differences between different CNTs (such as outer-wall vs. inner-wall CNTs), Figure R2d convincingly suggests that in practice they are not large enough to change the conclusion of our manuscript.

Figure R2. (a) Optical transition energies based on the empirical fitting functions as a function of diameter for semiconducting CNTs⁷. Shaded area indicates the diameter range for outer-wall CNTs determined by transmission electron microscopy (TEM). (b) (top) Datapoints from (a) corresponding to the outer-wall diameter range were plotted with reversed axis. (b) Absorbance spectrum of the annealed film for E_{11}^S range. Data was extracted from Supplementary Figure 7. (c) Schematic image of density of states (DOS) for semiconducting CNTs. E_{11}^S transition happens between the first sub-band of valence and conduction bands. Black dash line indicates an estimated position of Fermi energy (E_F). (d) Absorbance spectrum of the as-produced film (black solid line). Data was extracted from Supplementary Figure 7. (e) E_{11}^S transition is not allowed due to Pauli blocking when E_F is inside the valence band.

This comment helped us realize that we did not make this point clear in our previous manuscript, since details were only discussed in the Supplementary Information. **Hence, we have added the following sentences to the main text (Page 9, line 157-162)**, where we clarify how we estimated E_F based on optical spectroscopy measurements.

We estimated E_F through optical absorption spectral analysis. For example, the peak at ~0.57 eV is due to the E_{11} transition of the outer semiconducting CNTs; it is visible in the annealed samples but is suppressed in the as-produced and ICI doped samples. The suppression of the peak suggests that E_F resides inside the valence band, causing Pauli blocking. By analyzing other peaks in the same manner, the E_F of the annealed (as-produced and doped) samples was estimated to be in the vicinity of the first VHS of the outer (inner) semiconducting tubes (Supplementary Figure 7b).

Comment #2:

ZT needs to be plotted in the main manuscript so that readers can have the information without additional calculation.

Response to comment #2: We agree that ZT is an important quantity, and thus, this information should be included in the main text. While we did not measure the thermal conductivity of different samples under different conditions in this work (and thus we are unable to plot ZT), we estimated ZT for the CNT fiber with the largest power factor (annealed at 500 °C) by using the previously reported thermal conductivity value for a similar sample.⁸ **We have added the following sentence to the main text (Page 7, line 122-124).**

Assuming the thermal conductivity for similar annealed CNT fibers¹⁶, 580 Wm⁻¹K⁻¹, the average (maximum) ZT is estimated to be 6×10^{-3} (7×10^{-3}) at 300 K.

Comment #3:

1. LED has been turned on with 83 mV but I wonder if there is a LED that can be operated with such a low voltage. It is necessary to describe the specification of the LED.
2. Also the LED was turned on for a very short period of time while TE is supposed to generate electricity continuously. This would be possible presumably with the circuitry. More detailed information is necessary for readers to have a better understanding about this work.

Response to comment #3:

1. The reviewer is correct that an LED cannot be driven by a voltage as small as 83 mV. Indeed, there was nothing special about the LED (TEG-DMO, Custom Thermoelectric, LLC). We utilized a bootstrap converter (VB0410-1, TXL Group, Inc.), as mentioned in Page 13 line 249 in our previous manuscript. This DC/DC converter operates when the input voltage is 40 mV or higher, and the output voltage is 1 to 10 V, depending on the input voltage and load.
2. The LED was driven by a voltage that was discharged from capacitors. As we discussed in the Methods section (Page 24 line 405 – Page 25 line 407) as well as in the Supplementary Movie 1, we first charged the capacitors with the textile TE generator, and then discharged them to turn on the LED. In an RC discharging circuit, the voltage across the capacitor (V_c) at any instant in time during the discharging period is

$$V_c = V_s \exp\left(-\frac{t}{\tau}\right),$$

where V_s is the initial voltage across the capacitor, t is the time, and τ is the time constant, defined as $\tau = RC$, where R is the resistance and C is the capacitance. This equation indicates that V_c is 98.2 % discharged when $t = 4\tau$. In our case, the time constant was 0.61 seconds. The converter requires a minimum voltage of 40 mV to operate; otherwise, it outputs 0 V. Therefore, even when the initial voltage V_s is 80 mV, V_c is less than 40 mV after 0.43 seconds, which explains why the LED was on for a short period of time.

We agree that we did not make these points clear in our previous manuscript, since details were only discussed in the Methods section and in the Supplementary Movie 1. **Hence, we have added the following sentences to the main text (Page 13, line 249 – Page 14, line 254) as well as to the Methods section (Page 25, line 409 – Page 26, line 416), where we explain the operation principles of the converter and RC charging/discharging circuits. We have also changed the Supplementary Figure 20 to clarify that we utilized a DC/DC converter.**

The main text (Page 14, line 250-255)

Using this CNT-based textile TE generator, we demonstrated powering of a LED. The entire device consisted of sixty CNT threads connected, and we utilized RC circuits and a DC/DC converter to drive the LED, as shown in Supplementary Figure 20. We first charged the capacitors with the textile TE generator, and then discharged them to turn on the LED (see also Supplementary Movie 1). With a ΔT of >50 K, the textile TE generator provided a high enough voltage to light up the LED after the conversion, as shown in Figure 3e.

The Methods section (Page 25, line 411 – Page 26, line 418)

The converter operates when the input voltage is 40 mV or higher, and the output voltage is 1 to 10 V depending on the input voltage and load. When the switch is connected to A, the capacitor is charged by the TEGs through $Q(t) = CV_s[1 - \exp(-t/R_sC)]$, where Q is the charge, C is the capacitance, V_s is the voltage generated by the TEGs, t is time, and R_s is the resistance of the TEGs. When it switches to B, the capacitor discharges through $Q(t) = CV_C \exp(-t/R_LC)$, where V_C is the voltage drop across the capacitor at $t = 0$ and R_L is the resistance of the converter, to drive the LED (Supplementary Movie 1).

References used in responses

1. Hayashi, D. *et al.* Thermoelectric properties of single-wall carbon nanotube networks. *Jpn. J. Appl. Phys.* **58**, 075003 (2019).
2. Romero, H. E., Sumanasekera, G. U., Mahan, G. D. & Eklund, P. C. Thermoelectric power of single-walled carbon nanotube films. *Phys. Rev. B* **65**, 1–6 (2002).
3. Esfarjani, K., Zebarjadi, M. & Kawazoe, Y. Thermoelectric properties of a nanocontact made of two-capped single-wall carbon nanotubes calculated within the tight-binding approximation. *Phys. Rev. B* **73**, 085406 (2006).
4. Shiraishi, M. & Ata, M. Work function of carbon nanotubes. *Carbon* **39**, 1913–1917 (2001).
5. Shan, B. & Cho, K. First principles study of work functions of single wall carbon nanotubes. *Phys. Rev. Lett.* **94**, 236602 (2005).
6. Barone, V., Peralta, J. E., Uddin, J. & Scuseria, G. E. Screened exchange hybrid density-functional study of the work function of pristine and doped single-walled carbon nanotubes. *J. Chem. Phys.* **124**, 024709 (2006).
7. Weisman, R. B. & Bachilo, S. M. Dependence of optical transition energies on structure for single-walled carbon nanotubes in aqueous suspension: An empirical Kataura plot. *Nano Lett.* **3**, 1235–1238 (2003).
8. Behabtu, N. *et al.* Strong, Light, Multifunctional Fibers of Carbon Nanotubes with Ultrahigh Conductivity. *Science*. **339**, 182–186 (2013).

References used in transcribed sections from the revised main text

(reference numbers are identical to the ones in the revised main text)

16. Behabtu, N. *et al.* Strong, light, multifunctional fibers of carbon nanotubes with ultrahigh conductivity. *Science* **339**, 182–186 (2013).

REVIEWERS' COMMENTS

Reviewer #3 (Remarks to the Author):

The authors have made an acceptable level of responses to most of the comments. One last thing is to identify the power consumption of the boost converter, and also clarify if the boost converter was operated by the CNT thermoelectric device because the power required to operate the voltage booster should be supplied by the CNT device.

Response to Reviewer 3's Comments

We thank the reviewer for making valuable final comments. In the following, we respond to each of the comments:

Comment #1:

1. One last thing is to identify the power consumption of the boost converter,
2. and also clarify if the boost converter was operated by the CNT thermoelectric device because the power required to operate the voltage booster should be supplied by the CNT device.

Response to comment #1:

1. According to the performance curves for the bootstrap converter VB0410-1 provided by TXL Group, Inc. (<https://customthermoelectric.com/elc-uvb040-unipolar-voltage-booster-40mv.html>), a power consumption is 0.1 – 0.5 mW when the voltage provided by a thermoelectric power generator is 40 – 80 mV.
2. Yes, the converter was solely operated by the CNT TEGs and no additional power was required. **We have modified the Methods section (Page 29, line 538-540) to clarify this point. We have also changed the Supplementary Figure 20 to avoid any confusion about the converter.**

The Methods section (Page 29, line 538-540)

When it switches to B, the capacitor discharges through $Q(t) = CV_C \exp(-t/R_L C)$, where V_C is the voltage drop across the capacitor at $t = 0$ (thus $V_C = V_S$) and R_L is the resistance of the converter, to drive the LED (Supplementary Movie 1). Note that the converter was solely operated by the TEGs and no additional power was required.